

# Towards an efficient storm surge and inundation forecasting system over the Bengal delta: Chasing the super-cyclone Amphan

Md Jamal Uddin Khan[1], Fabien Durand[1,2], Xavier Bertin[3], Laurent Testut[1,3], Yann Krien[3], A.K.M. Saiful Islam[4], Marc Pezerat[3], and Sazzad Hossain[5,6]

[1]LEGOS UMR5566, CNRS/CNES/IRD/UPS, 31400 Toulouse, France
[2]Laboratório de Geoquímica, Instituto de Geociencias, Universidade de Brasilia, Brazil
[3]UMR 7266 LIENSs, CNRS- La Rochelle University, 17000 La Rochelle, France
[4]IWFM, BUET, Dhaka 1000, Bangladesh
[5]Flood Forecasting and Warning Centre, BWDB, Dhaka, Bangladesh
[6]Department of Geography and Environmental Science, University of Reading, UK

**Correspondence:** Md Jamal Uddin Khan (jamal.khan@legos.obs-mip.fr)

**Abstract.** The Bay of Bengal is a well-known breeding ground to some of the deadliest cyclones in history. Despite recent advancements, the complex morphology and hydrodynamics of this large delta and the associated modelling computational costs impede the storm surge forecasting in this highly vulnerable region. Here we present a proof of concept of a physically consistent and computationally efficient storm surge forecasting system tractable in real-time with limited resources. With a

state-of-the-art wave-coupled hydrodynamic numerical modelling system, we forecast the recent super cyclone Amphan in real-time. From the available observations, we assessed the quality of our modelling framework. We affirmed the evidence of the key ingredients needed for an efficient, real-time surge and inundation forecast along this active and complex coastal region. This article shows the proof of the maturity of our framework for operational implementation, which can particularly improve the quality of localized forecast for effective decision-making.

## 1 Introduction

Storm surges and associated coastal floodings are one of the most dangerous natural hazards along the world coastlines. Annually, storm surges killed on average, over 8000 people and affected 1.5 million people worldwide over the past century (Bouwer and Jonkman, 2018). Among the storm surge-prone basins, the Bay of Bengal in the northern Indian Ocean is one of the deadliest. This basin has consistently been home to the world's highest surges, experiencing in each decade an average of

five surge events exceeding 5 m (Needham et al., 2015). On May 18th of 2020, a super-cyclone named Amphan was identified as the most powerful ever recorded in the Bay of Bengal, with highest sustained 3-min wind speed more than 240 km per hour (130 knots) and highest 1-min wind gusts as fast as 260 km/h (https://www.metoc.navy.mil/jtwc/jtwc.html). Two days later, on May 20th, Amphan struck the coasts of Bangladesh and India, with a sustained wind speed of 112 km per hour and gusts of 190 km per hour, causing massive damage and claiming hundreds of lives.

The cyclone activity in the Bay of Bengal is very different from the other cyclone-prone oceanic basins (Needham et al., 2015). The basin experiences a distinct bi-modal distribution of cyclonic activity - one with a peak in the pre-monsoon (April-





May) and another during post-monsoon (October-November) (Bhardwaj and Singh, 2019). The Bay of Bengal is a small semi-enclosed basin (Figure 1), which accounts for only 6% of cyclones worldwide. However, more than 70% of global casualties from the cyclones and associated storm surges over the last century occurred there (Ali, 1999). The number of fatalities

concentrates in the Bengal delta across Bangladesh and India (Ali, 1999; Murty et al., 1986) where more than 150 million people live below 5 m above mean sea level (MSL) (Alam and Dominey-Howes, 2014). Seo and Bakkensen (2017) noted a statistically significant correlation between storm surge height and the fatality in this region. Two of the notable cyclones that struck this delta include the 1970 cyclone Bhola and 1991 cyclone Gorky which killed about 300,000 (Frank and Husain, 1971) and 150,000 (Khalil, 1993) people respectively. In recent decades, the death toll has reduced by orders of magnitude. For

example, cyclone Sidr, a category five equivalent cyclone on the Saffir-Simpson scale that made landfall in 2007, claimed 3406 lives (Paul, 2009). At the same time, the cost of material damages has significantly increased (Alam and Dominey-Howes, 2014; Emanuel, 2005; Schmidt et al., 2009). During recent cyclones, government and voluntary agencies of Bangladesh and India took coordinated effort to evacuate millions of people to safety at cyclone shelters before the cyclone landfall (Paul and Dutt, 2010). This kind of informed coordination benefited from improved communication, increased shelter infrastructure, and,

most importantly, from the improvement of the numerical prediction of storm track and intensity.

Over the last decades, global weather and forecasting systems have advanced significantly. Several global models now run multiple times a day, starting from multiple initial conditions, with horizontal resolution in the order of tens of km, providing forecast with a range of hours to a week. These forecasts have proven useful during weather extremes like tropical storms (Magnusson et al., 2019). Operational hurricane forecasting systems like Hurricane Weather Research and Forecasting (HWRF)

have emerged and reached a level of maturity to provide reliable cyclone forecasts several days in advance throughout the tropics (Tallapragada et al., 2014).

Storm surge forecasting has been developed and implemented around the world along with the advancement of the weather and storm forecasting (Bernier and Thompson, 2015; Daniel et al., 2009; Flowerdew et al., 2012; Lane et al., 2009; Verlaan et al., 2005). Nowadays, operational surge forecasting systems typically run on high-performance computing systems, either

on a scheduled basis or triggered on-demand during an event (Khalid and Ferreira, 2020). In the Bay of Bengal region, the Indian Institute of Technology-Delhi (IIT-D) storm surge model has been running operationally with a horizontal resolution of 3.7 km for one decade (Dube et al., 2009). Under the Tropical Cyclone Program (TCP) of the World Meteorological Organization (WMO), IIT-D model is also implemented to run in Bangladesh Meteorological Department (BMD). Beside the IIT-D model, BMD also experimentally uses MRI storm surge model from Japan Meteorological Agency (JMA) (http:

50    //bmd.gov.bd/p/Storm-Surge). Storm surge forecasts have shown their potential to better target the evacuation decision, to optimize early-engineering preparations, and to improve the efficiency of the allocation of the resources (Glahn et al., 2009; Lazo and Waldman, 2011). Keeping in mind the cyclonic surge hazards over the densely populated Bengal delta, having a reliable real-time operational forecast system in the region would be extremely valuable.

Storm surge modelling is, however, computationally expensive and proved to be challenging in real-time forecasting mode

(Glahn et al., 2009; Murty et al., 2017). Several practical solutions exist to curb the real-time constraints of cyclone forecast, such as soft real-time computation based on looking up an extensive database of pre-computed cyclones (Condon et al., 2013;


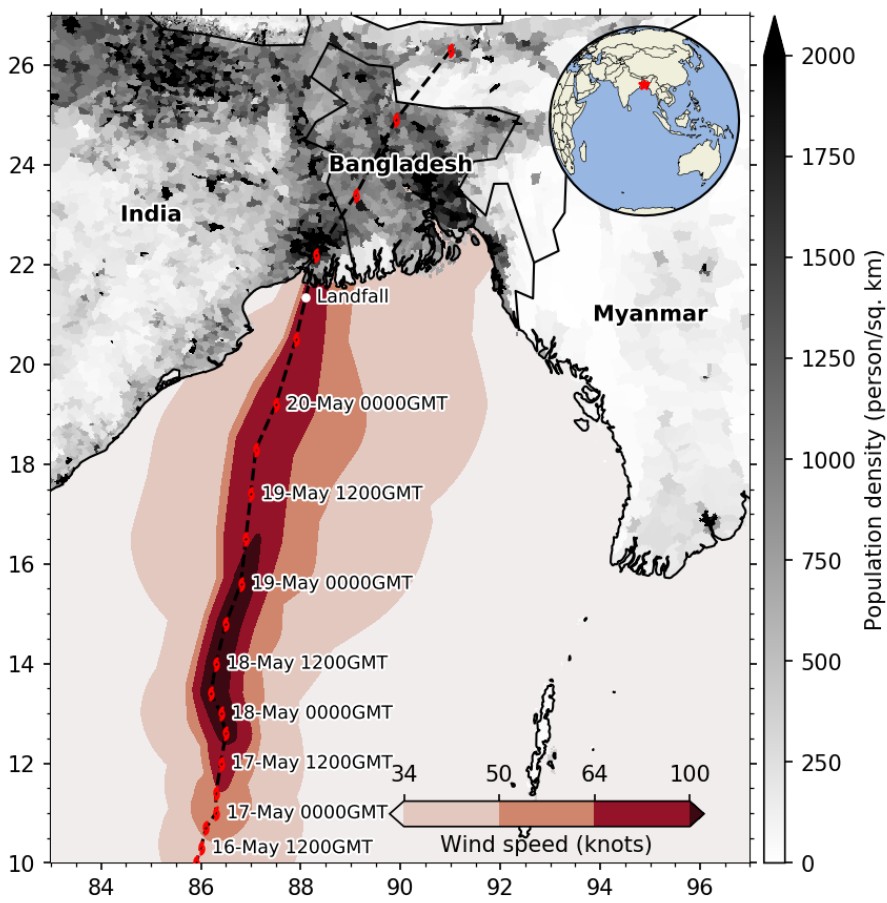

**Figure 1.** The path of the super cyclone Amphan of May 2020, overlaid on the population density. The footprint of 34 knots (17.5 m/s), 50 knots (25.7 m/s), 64 knots (32.9 m/s), and 100 knots (51.4 m/s) wind speed area is shown with the red colour bar. Inset shows the location of the study area.

Yang et al., 2020), coarse-resolution modelling (Suh et al., 2015), and reduced-physics modelling (Murty et al., 2017). In the past decade, unstructured-grid modelling systems are getting more and more popular due to their efficiency in resolving the topographic features and their reduced computational cost compared to structured-grid equivalents (Ji et al., 2009; Lane et al., 2009; Melton et al., 2009).

The published history of storm surge modelling in the northern Bay of Bengal goes hand-to-hand with the land-falling of very severe cyclones (Das, 1972; Flather, 1994; Dube et al., 2004; Murty et al., 2014; Krien et al., 2017). From the modelling of the historical storm events, previous studies have identified several ingredients as essential for accurate modelling and forecasting of storm surge induced water level and associated flooding over the Bengal delta. The most important of these ingredients is an accurate bathymetry and topography (Krien et al., 2016, 2017; Murty et al., 1986). Second, it is required to have a large-

scale modelling domain comprising the whole Ganges-Brahmaputra-Meghna (GBM) estuarine network (Johns and Ali, 1980;





Oliveira et al., 2020) at a high-enough model resolution (Krien et al., 2017; Kuroda et al., 2010). The modelling framework is required to simulate tide and surge together to account for the tide-surge interactions (As-Salek and Yasuda, 2001; Johns and Ali, 1980; Kuroda et al., 2010; Murty et al., 1986). Furthermore, an online coupling of the hydrodynamics and the short waves is also necessary to account for the wave setup (Deb and Ferreira, 2016; Krien et al., 2017). The capability of the model to simulate the wetting and drying is also necessary to model the inundation (Madsen and Jakobsen, 2004). However, due to limited computing resources, storm surge models typically used in the Bay of Bengal region for forecasting purpose are coarse, without any coupling of the hydrodynamics with the waves, and occasionally uncoupled to tide (Dube et al., 2009; Murty et al., 2017, 1986; Roy et al., 2015). These limitations in operational forecasting systems can significantly impede a proper interpretation during an actual cyclonic storm, as discussed in the next section through a set of hindcast sensitivity experiments of cyclone Amphan.

Cyclone Amphan is not only the latest event on record in the Bay of Bengal but also the costliest event that struck this shoreline, with an estimated bill over 14 billion dollars in West Bengal and Bangladesh (DhakaTribune, 2020; IFRC, 2020). During this cyclone, we have proactively forecast storm surge in real-time with common computational resources using a high-resolution coupled modelling system forced by a combination of freely available atmospheric forecasts. The general goal of this paper is to provide a proof-of-concept of a tractable operational storm surge forecasting system over this key-region of vulnerability, to identify the key ingredients of such a system, and to provide guidance in the near-future initiatives of the operational forecasting community. First, we present the various processes governing the surge dynamics to illustrate the challenges of modelling storm surges in the Bay of Bengal in Section 2. Section 3 documents the atmospheric forecasts that are required to generate a surge forecast. In Section 4, we introduce our numerically-efficient hydrodynamics-waves coupled modelling platform and present its practical real-time computational setup in Section 5. Finally, we present the remaining modelling and forecasting issues in Section 6. Section 7 provides a conclusion to this study.

## 2 Storm surge and inundation processes in the Bay of Bengal

The Bay of Bengal is a macro-tidal region where peak tidal range reaches as high as 5 m over the north-eastern corner of the Bay (Krien et al., 2016; Tazkia et al., 2017). A large geographical extent of the land-sea hydraulic continuum with an intricate estuarine network complicates the water level dynamics in this part of the coastal ocean. During a cyclone, the dynamics get even more complicated due to wave-induced setup and tide-surge interaction. In Figure 2, we illustrate a simplified view of water level components – tide, surge, wave setup – and their interactions with the topography in driving inundation during a cyclone.

During a cyclone, the atmospheric pressure drop and the wind stress both generate the water level surge. This atmospheric storm surge is non-linearly dependent on the astronomical tide. The dependent nature of both tide and surge implies that the surge induced by a given meteorological condition differs at different stages of the tide, particularly in the nearshore shallow zone. This departure from linearly-added tide and surge component is known as tide-surge interaction. In a macro-tidal region like the Bengal delta tide-surge interaction typically amounts to several tens of centimetres (Antony and Unnikrishnan, 2013;





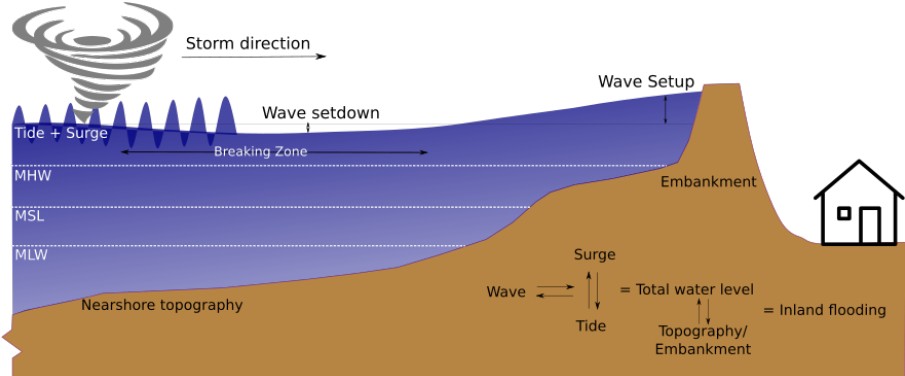

**Figure 2.** Conceptual diagram of the involved processes that determine the water level evolution and its interaction with the controls determining the inland flooding.

Idier et al., 2019). Due to this interaction, the highest surge is obtained for a storm making landfall around 2 hours before the high tide. The strong dependence on tide reinforces the importance of having an accurate tidal model for the region and a reliable network of water level gauge for validation (As-Salek and Yasuda, 2001; Krien et al., 2017; Murty et al., 2016).

Wave setup is another component that has a significant impact on the nearshore sea level (Idier et al., 2019; Krien et al., 2017; Murty et al., 2014). It manifests as an increase in the sea level occurring in the nearshore zone that accompanies the dissipation of short waves by breaking (Longuet-Higgins and Stewart, 1962). During cyclone Sidr, the modelled wave-setup was around 30 cm (Krien et al., 2017). This estimation would be potentially underestimated by up to 100% due to an early dissipation of wave energy by depth-induced breaking arising from usual parameterizations utilized in spectral wave models Pezerat et al. (2020).

At the seasonal scale, the mean sea level around the Bengal delta also shows considerable evolution. The amplitude of this variation can go as high as 40 cm due to freshwater influxes during monsoon from the GBM river system and offshore-ocean steric variability (Durand et al., 2019; Tazkia et al., 2017). During the wintertime cyclone-prone season, this steric variability can induce 10-15 cm variation of mean sea level of the bay.

Except for the Sundarban mangrove forest, coastal Bangladesh is mostly embanked through a network of 139 polders (Nowreen et al., 2013). These embanked areas can be flooded in three ways - (a) by overflowing/overtopping, (b) by breaching of the embankments/control structures, and (c) by heavy rainfall inside the embanked area. The height of these embankments plays the most vital role in actuating the inundation from a storm surge (Krien et al., 2017). The performance of the embankments during the cyclone is another crucial factor, depending on the composition (typically, earth vs stones/concrete). During a cyclone, the wave action as well as overtopping/overflowing on the embankments can create breaches at a weak point, thus creating local inland flooding (Islam et al., 2011). Particularly for Bangladesh, the consequence of such storm surge induced flooding can be long-lasting as the topography inside polders is often below the mean sea level due to ground compaction (Auerbach et al., 2015).





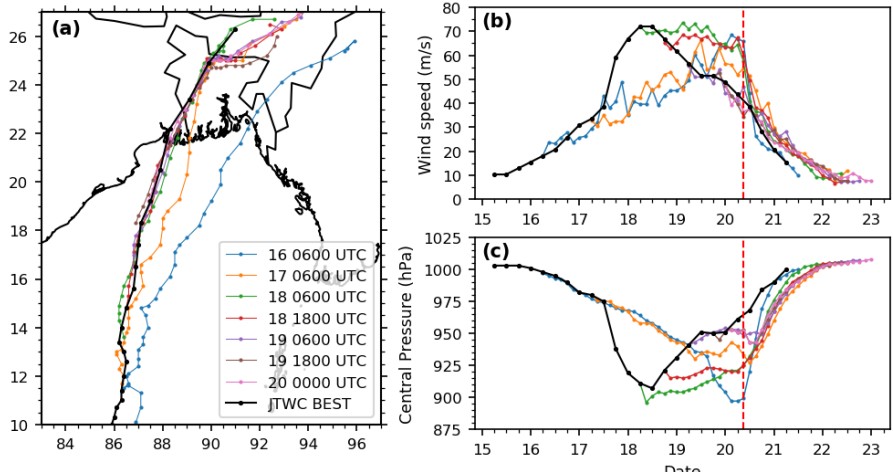

**Figure 3.** Temporal evolution of the successive forecasts of Amphan cyclone wind and pressure. (a) Forecast track colour-coded with the date (JTWC best track in black) (b) Wind speed forecast with each epoch, (c) Pressure forecast with each epoch. The vertical red dashed line indicates the time of landfall.

## 3  Atmospheric evolution of cyclone Amphan

On May 13th of 2020, a low-pressure area was persisting over northern Bay of Bengal about 300 km east of Sri Lanka. By the end of 15th, the Joint Typhoon Warning Center (JTWC) upgraded the low-pressure system to a tropical depression. Indian
Meteorological Department (IMD) also reported the same development on the next day. The tropical depression continued to move northward and intensified into the cyclonic storm Amphan by May 16th 18:00 UTC. During the following 12 hours, the intensification of the system was limited. However, starting from 12:00 UTC on May 17th, Amphan started to intensify very rapidly. In just 6 hours around 18:00 UTC, the maximum wind speed increased from 140 km/h to 215 km/h, making it an extremely severe cyclonic storm (equivalent to Category 4 on the Saffir-Simpson scale). Over the next twelve hours,
Amphan continued to intensify reaching a maximum of 260 km/h wind-speed and 907 millibars central pressure, making it the most intense event on record in the Bay of Bengal. During this time, it appeared to form two distinct eye-walls, which is typical of intense cyclones. Over the next 24 hours, Amphan lost most of its strength due to the eye-wall replacement cycle in the presence of moderate vertical wind shear (30-40 km/h). The system continued to decay due to easterly shear and mid-level dry air. The vertical wind shear remained moderate during this period. The cyclone made landfall between 08:00UTC
and 10:00UTC around (88.35°N,21.65°E) (Figure 1), at mid-tide. At landfall, the reported central pressure amounted to 965 millibars with a maximum wind speed of 150-160 km/h. Afterwards, over its journey inland, the system further eroded and disappeared by May 21st. The black lines in Figure 3 illustrates the evolution of Amphan from JTWC best track.

The extended range outlook by IMD published on May 7th predicted cyclogenesis to occur during May 8th to May 14th with low probability RSMC (2020). Global and regional models started to predict a significant storm to happen in the Bay of Bengal





as early as May 12th, eight days before the cyclone landfall and four days before the actual formation of the tropical depression. The formation of the low-pressure system triggered the operational HWRF system of NOAA (https://www.emc.ncep.noaa.gov/gc_wmb/vxt/HWRF/) on May 14th. Similarly, the operational system of IMD (https://nwp.imd.gov.in/hwrf/IMDHWRFv3.6) was also triggered a day later. For the rest of the paper, we confine our analysis to the forecast disseminated by NOAA-HWRF only, relying on the Automated Tropical Cyclone Forecasting System (ATCF) text output (Miller et al., 1990). Figure

3 illustrates a few of the selected forecast cyclone tracks sequentially issued from NOAA-HWRF.

  The forecasts illustrate the convergence of the location of the landfall. As early as May 17th, three days before the landfall, the forecast tracks had converged towards the observed track. One day later, on May 18th, the forecast error on the storm track location had reduced to around 50 km. The forecast wind speed and central pressure did not show as much accuracy as the forecast trajectory. The initial forecasts captured the magnitude of the intensification. However, they failed to forecast the rapid

intensification that occurred during the 17-18th of May. On the other hand, the subsequent forecasts initiated during May 17th and May 18th failed to capture the rapid weakening of the system that occurred over May 19th-20th, before landfall. Overall, in terms of wind speed, the forecast evolution was accurate (within 1 m/s) only at 24 hours range.

## 4 Storm surge model and performance

We have developed our tide-storm surge model based on the community and open-source modelling platform SCHISM (Semi-

155 implicit Cross-scale Hydroscience Integrated System Model) developed by Zhang et al. (2016b) - a derivative code of SELFE (Semi-implicit Eulerian-Lagrangian Finite Element) model, originally developed by Zhang and Baptista (2008). This model solves the shallow-water equations using finite-element and finite-volume schemes in an unstructured grid that can combine tri- and quad- elements. The model is applicable in baroclinic as well as barotropic ocean circulation problems, for a broad range of spatial scales from the creek scale to the ocean basins (Ye et al., 2020; Zhang et al., 2020, 2016a). It has already been shown

to have an excellent performance in reproducing shallow water processes over the Bengal shoreline and elsewhere, including the coastal tide (Krien et al., 2016), wave setup (Guérin et al., 2018) and storm surge flooding (Bertin et al., 2014; Krien et al., 2017; Fernández-Montblanc et al., 2019).

### 4.1 Model implementation

We have implemented our model on an updated bathymetry of Krien et al. (2016) with the additional inclusion of 77,000

digitized sounding points from a set of 34 nautical charts published by Bangladesh Navy (Khan et al., 2019). Our bathymetric dataset is a blend of two digitized sounding datasets in the nearshore zone – one from navigational charts produced by Bangladesh Navy, and another being a bathymetry of the Hooghly estuary provided by IWAI (Inland Waterways Authority of India). The river bathymetry is composed of a set of cross-sections obtained from the Bangladesh Water Development Board (BWDB). The south-central part of the delta is composed of a high-resolution (50 m) inland topography. We considered

GEBCO 2014 bathymetry to complement in the deeper part of the ocean (https://www.gebco.net/data_and_products/gridded_bathymetry_data/), and SRTM digital elevation model for the rest of the inland topography as appearing in the GEBCO 2014



dataset (https://www2.jpl.nasa.gov/srtm/cbanddataproducts.html). Our unstructured model mesh was developed based on this blended bathymetry, covering the whole Bay of Bengal (11°N to 24°N) with a variable resolution based on the shallow water wave propagation and bottom slope criteria. Our model grid consists of about 600,000 nodes and 1M elements in total. The resulting grid resolution ranges from 250 m in the estuaries and the delta, to 15 km in the deeper part of the central Bay of Bengal. The model domain and mesh are identical to those presented in Khan et al. (2019) (see their Figure 9).

To account for the effect of short waves on the mean circulation, Wind Wave Model III (WWMIII), a third-generation spectral wave model, is coupled online with SCHISM in our modelling framework (Roland et al., 2012). In our configuration, WWMIII solves wave action equation on the native SCHISM grid, using a fully implicit scheme. The source terms in our model comprise the energy input due to wind (Ardhuin et al., 2010), the non-linear interaction in deep and shallow water, energy dissipation in deep and shallow water due to white capping and wave breaking, and energy dissipation due to bottom friction. We run WWMIII with a 12 directional and 12 frequency bins. Water level and current are exchanged among the two models every 30 minutes. Wave breaking is modelled, according to Battjes and Janssen (1978). As the nearshore region of Bengal delta has a very mild slope, the value for the breaking coefficient $\alpha$, which controls the rate of dissipation, was reduced from 1 to 0.1 to avoid over-dissipation (Pezerat et al., 2020). WWMIII was also forced along the model southern open boundary by time series of directional spectra, computed from a large-scale application of the WaveWatch3 (hereafter WW3) model (WW3DG, 2019). WW3 was implemented at the scale of the whole Indian Ocean with a spatial resolution of 0.5°and forced by the FNL reanalysis wind data at 3-hour time step (National Centers for Environmental Prediction, National Weather Service, NOAA, U.S. Department of Commerce, 2015).

## 4.2   Model Forcings

Our model is forced over the whole domain by the astronomical tidal potential for the 12 main constituents (2N2, K1, K2, M2, Mu2, N2, Nu2, O1, P1, Q1, S2, T2). At the southern boundary, we have prescribed a boundary tidal water level from 26 harmonic constituents (M2, M3, M4, M6, M8, Mf, Mm, MN4, MS4, Msf, Mu2, N2, Nu2, O1, P1, Q1, R2, S1, S2, S4, Ssa, T2, K2, K1, J1, and 2N2) extracted from FES2012 global tide model (Carrère et al., 2013). At each of the upstream river open boundaries of Ganges, Brahmaputra, Hooghly, and Karnaphuli, we implemented a discharge boundary condition. At Meghna and Rupnarayan river open boundaries, we implemented a radiating Flather boundary condition. From a year-long tidal simulation, we found that the updated version of the bathymetry performs 2-4 times better at key locations compared to the global tidal models – which includes FES (Carrère et al., 2013), GOT (Ray, 1999), and TPXO (Egbert and Erofeeva, 2002) (see Table A1).

We derived the atmospheric wind and pressure fields for SCHISM from a blending of analytical wind field inferred from a parametric wind model (close to the cyclone track) and background atmospheric field from the Global Forecast System (GFS, https://www.ncdc.noaa.gov/data-access/model-data/model-datasets/global-forcast-system-gfs) reanalysis (further away), following Krien et al. (2017, 2018). We used the best-track data of JTWC provided at 6-hour time step as input for the analytical wind and pressure field. Here we choose the analytical wind profile of Emanuel and Rotunno (2011) and the analytical pressure field profile of Holland (1980). To be consistent with our forecast described in Section 5, the background wind field was gener-





ated incrementally from an accumulative merging of GFS forecasts for each 6-hour forecast window, with a 1-hour time step. The analytical and background wind fields were first temporally interpolated every 15 minutes and overlaid on the background GFS fields using a distance-varying weighting coefficient to the cyclone centre to ensure a smooth transition. We took into account the asymmetry of the cyclonic wind field following Lin and Chavas (2012).

## 4.3 Predictive skills

With the above-described model setup, we ran our model to calculate the water level and sea state evolution during Amphan. The simulation was done a-posteriori, once the cyclone had passed, and JTWC and GFS hindcasts being already available throughout the cyclone lifetime. Figure 4 shows comparisons between observed and modelled significant wave heights (SWH), peak wave period and water level. First, we compare the significant wave height derived from the altimetric estimate of Sentinel-3B processed by the Wave Thematic Assembly Center from Copernicus Marine Environment Monitoring Service (https://scihub.copernicus.eu/), which overpass on May 18th 2020 at 16:03GMT. For each altimetric data point inside our domain, we interpolated the model output using a nearest-neighbour approach, both spatially and temporally. This comparison shows that SWH is reasonably reproduced, with RMSE typically within 1m (within 18% of the mean value). However, we observed an overestimation of SWH of about 2.5 m near the cyclone centre. One of the reasons for such difference is a limitation of analytical wind field models, as discussed by Krien et al. (2017).

We present an in situ observed time-series of SWH and mean wave period during the life cycle of Amphan in Figure 4 for BD08 (Figure 4, b-c) and BD11 (Figure 4, d-e) oceanographic buoys (the dataset is available online at https://incois.gov.in/portal/datainfo/mb.jsp). The two buoys are located on either side of Amphan track, 250 km to its east / 230 km to its west, respectively. We can see that at both locations, the model reproduces the SWH well. The match is particularly good at BD11, consistently within 1 m of the observed evolution. At BD08, SWH is found to be overestimated by about 2 m at its peak, with a 12-h lead shift in time. Such difference can come from positional and timing uncertainties in JTWC best track data, from our assumption of a constant translational velocity in-between each 6-hourly JTWC time step, and finally from the intrinsic uncertainty of the analytical wind field itself, as shown on Figure 3b. Despite multiple sources of potential errors and uncertainties, our model appears capable of capturing the general features of the sea state, both significant wave height (SWH) and period, relatively well. The overall predictive skills of the model in terms of short waves are only slightly below those from hindcast exercises typically conducted over other ocean basins using similar modelling strategies (Bertin et al., 2014; Krien et al., 2018).

The real-time availability of observed water level time series in this region is relatively scanty. We were able to access only a handful of coastal water level records, most of them located eastward of the landfall location. The right panels of Figure 4(f-k) illustrated the comparison of the recorded water level and modelled storm surge. Among these tide-gauges, Chittagong (Figure 4j) and Visakhapatnam (Figure 4k) time series are retrieved from the UNESCO IOC Sea Level Monitoring service (http://www.ioc-sealevelmonitoring.org/map.php) and the rest are provided by Bangladesh Inland Water Transport Authority (BIWTA, http://www.biwta.gov.bd/). The datum for water level timeseries at Chittagong and Visakhapatnam is changed from in-situ datum to MSL by removing long-term mean. The timeseries provided by BIWTA are already referenced to MSL, thus

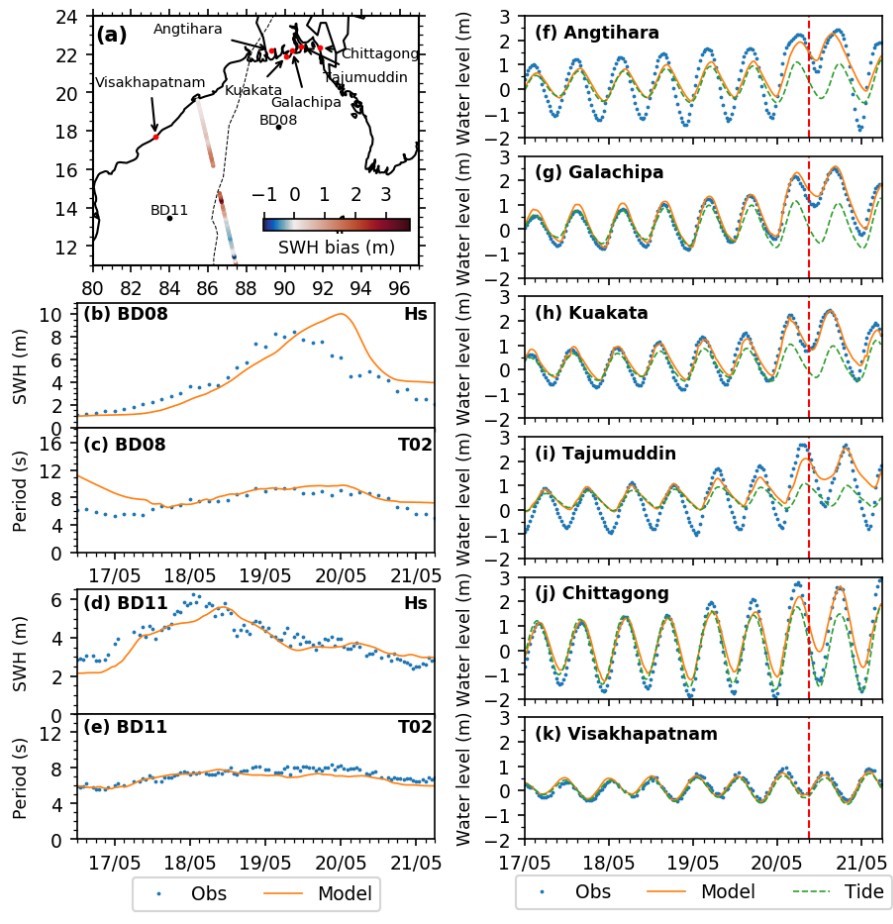

**Figure 4.** Comparison of simulated (in orange) and observed (in blue) water level, significant wave height (SWH) and mean wave period. (a) The map shows the along-track bias in SWH compared to the one calculated from Sentinel 3B altimeter overpass at 2020-05-18 1603Z. Bottom panel shows the modelled SWH and mean wave period (orange line) compared to buoy observations (blue dots) at BD08 (b-c) and BD11 (d-e) provided by INCOIS. Comparison between observed (blue dots) and modelled (orange line) water level at the station locations – (f) Angtihara, (g) Galachipa, (h) Kuakata, (i) Tajumuddin, (j) Chittagong, and (k) Visakhapatnam. Green dashed lines in (f)-(k) indicates the modelled tidal water level. Location of the buoys and the water level gauges are shown in (a). The vertical red lines in water level plots indicate the time of landfall.

kept unchanged. We also plotted the reconstructed tidal water level from the tidal atlas derived from our model. The overall tidal propagation is well simulated in Galachipa and Kuakata, while at Angtihara and Tajumuddin the simulation tidal range is found to be underestimated. The local bathymetric error and friction parameterization might be the source of the discrepancy. The tide gauge at Angtihara is located in a data-scarce location inside Sundarbans mangrove forest. Tide gauge at Tajumuddin is situated at the mouth of Meghna, where bathymetry is rapidly changing (Khan et al., 2019). However, our model correctly reproduces the peak of the water level at all locations. Depending on the site, the landfall occurred 2-4 h after high tide. The





maximum recorded water level at these stations is around 2-2.5 m above mean sea level, with a storm surge (water level – tide) of the same order of magnitude.

To spatially assess the storm surge generated by cyclone Amphan, we first take a look into the temporal maximum water level in Figure 5(a). We can see that the whole coastal region experienced a high-water level ranging from 2-5 m for Amphan.
To quantify the contribution of the cyclone, we have looked in the surge component, defined as the difference between the total modelled water level and a tide-only simulation. Figure 5(b) illustrates the temporal maximum of surge over the delta region. The highest surge level is 5 m around the location of the landfall (88.3°E, 21.6°N). The maximum non-linear interaction between tide and surge amounts to about 30 cm in the nearshore domain (Figure 5d).

To quantify the contribution of the wave setup, we re-calculated the storm surge without the coupling with the wave model.
Figure 5(c) shows the maximum of the difference between the two simulated water levels. In general, wave setup contributes to the Amphan storm surge by about 20 cm all along the Bengal shoreline, and locally over 30 cm close to the cyclone landfall. Nevertheless, the spatial resolution at the nearshore region employed in this study (250 m) is still too coarse for capturing the maximum wave setup that develops along the shoreline, where a resolution of a few tens of m should be employed (Guérin et al., 2018). Nonetheless, this comparison shows that wave setup is not only developed along the shoreline exposed to waves
but affects the whole delta up to far upstream, a process already observed over large estuaries (Bertin et al., 2015; Fortunato et al., 2017).

Our findings reaffirm the necessity of a proper coupling between tide, surge and wave to forecast the water level evolution. For Amphan, the magnitude of tide-surge interaction amounts to about 10% of the maximum total water level with spatial variation. Similarly, the magnitude of wave-setup is also spatially varying with a typical amplitude of 10-15% of the maximum
total water level. This non-negligible non-linear dependency shows the importance of a fully coupled hydrodynamic-wave modelling system for a successful forecast of the water levels in the hydrodynamic setting of the Bengal delta. Our modelling framework showed reasonable accuracy in reproducing observed storm surge water level along the Bengal coastline. The accuracy is in line with the typical level of performance of similar systems applied elsewhere in the world ocean (Bertin et al., 2015; Fernández-Montblanc et al., 2019; Suh et al., 2015). The hindcast experiment thus shows the viability of our model for
a proof-of-concept of real-time forecasting exercise.

## 5   Near real-time storm surge forecasting

The current state of maturity of atmospheric real-time cyclone forecast products allows their application to real-time surge forecasting, as discussed in Section 3. As explained in Section 2, the storm surge generated from cyclones depends on the atmospheric pressure, wind, and background (typically: tidal) water level. A realistic surge forecast issued three, or even two
275    days before landfall can usefully predict the order of magnitude of the expected maximum water level. Particularly a spatial prediction of expected maximum water level could be of utmost utility to determine the locations of impact and associated damages. This two to three-day timeframe can help to prepare the embankments or critical systems over the vulnerable areas, and prepare for the evacuation of the population living in low-lying coastal zones. Storm surge and water level evolution



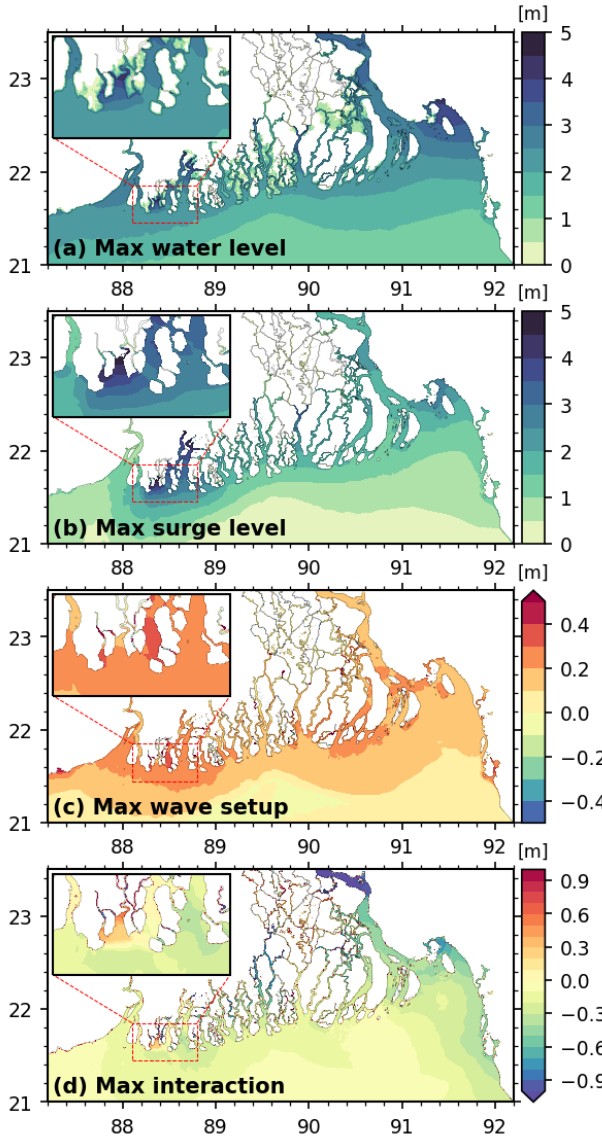

**Figure 5.** Hindcast of (a) maximum water level, (b) maximum surge (c) wave setup/setdown (d) maximum non-linear interaction between tide and surge. For (a), for the areas above mean sea level, the water level is converted to water level above the topography for consistency. The inset maps show a close-up (75 km × 45 km) of the landfall region.

estimated 24 hours to 12 hours before landfall, once the cyclone forecasts from the operational agencies have converged, can
further help to elaborate location-specific evacuation orders.





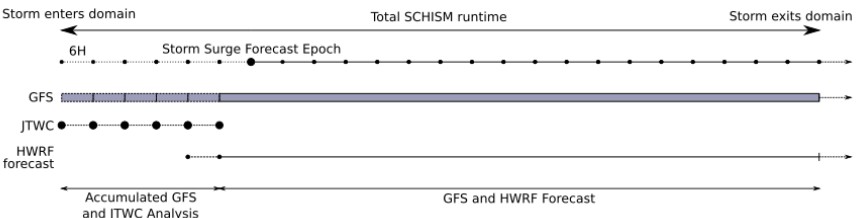

**Figure 6.** Temporal combination scheme of the JTWC, GFS, and HWRF forecasts for each 6-hourly storm surge forecast epoch.

## 5.1 Forecast strategy and forcing products

During cyclone Amphan, we performed near-realtime storm surge forecasts based on our model. We communicated the results to Bangladesh local government authority through personal communications, as well as to the scientific community through social media. In our forecasts, we relied on the outputs of global models (GFS, and HWRF) and advisories (JTWC) for the

285 estimates of the atmospheric forcing. These advisories and global models are updated every 6 hours. Similar to the GFS forecast cycle, we updated our forecasts at every 00:00, 06:00, 12:00, 18:00 UTC.

For each forecast, we derived the atmospheric forcing from a blend of JTWC, HWRF, and GFS data (Table A2). JTWC publishes storm advisories at 03:00, 09:00, 15:00, 21:00 UTC. The advisory includes an analysis of the storm intensity and position from a satellite fix 3-hours prior, followed by a forecast for the following 72 hours, with 6 to 12 hours time-steps.

NOAA published their HWRF model advisory for each GFS forecast cycle at 3-hour time steps. GFS provides the wind and pressure fields in hourly timesteps at 0.25°regular spatial grid. The GFS model is initialized 6-hourly at 00:00, 06:00, 12:00, 18:00 UTC, and the data is available 3 hours after the initialized period.

The wind and pressure fields around the centre of the storm are derived analytically from a concatenated JTWC and HWRF. We used GFS data as the background wind and pressure on the outer region of the analytical fields.

We initiated our forecast cycles on March 16th 2020 at 06:00 UTC - utilizing the first advisory from JTWC published at 03:00 UTC, merged with the forecast from HWRF issued at 18:00 UTC of the preceding day. We took the background wind and pressure field from GFS forecast published at 00:00 UTC. In the subsequent forecast cycles, we updated the previous track file by first replacing the past time-steps with the analysis from the latest JTWC advisory, then appending the forecasts from HWRF again. For the background wind and pressure fields from GFS, we retained the first 6 hours of forecasts from the

previous cycles and updated the remainder of the time series from the latest forecast with new initialization. For each forecast cycle, we kept the starting time of our model the same, on May 16th at 00:00 UTC, and ran till one-day after the landfall for a consistent initial condition. This approach is feasible as the storms in the Bay of Bengal typically form, grow and dissipate in about a week. A graphical representation of our strategy to temporally concatenate JTWC, GFS, and HWRF forecasts over a forecast cycle is shown in Figure 6.



**Table 1.** Computing environment used during the forecast

| | |
|---|---|
| Model simulation duration | 5.5 days |
| Model timestep | 5min |
| WWM Coupling Timestamp | 30min |
| Processor clock speed | 2.8GHz |
| Number of MPI processes | 20 |
| Wall clock time (pre-processing) | 10min |
| Wall clock time (model integration) | 1h45min |
| Wall clock time (post-processing) | 5min |
| Disk storage (each simulation) | 8GB |
| Memory usage | 18GB |

## 5.2 Real-time computation and results

Since our forecast is contingent to the JTWC advisory of three hours before, we had only a three-hour-long window to achieve the pre-processing of the forcing files, to run the model, and to complete the necessary post-processing of the model outputs. The numerical efficiency of our model made it possible to cope with this time constraint of the forecast, using a modest computing resource. The computing environment we used is detailed in Table 1, which essentially amounts to a single desktop computer fitted with a high-end consumer-grade processor.

Figure 7 illustrates the evolution of the maximum total water level and surge for the corresponding forecasts issued at T-60 hours, T-36 hours, and T-12 hours, where T is the time of landfall (May 20th 2020 at 08:00 to 10:00 UTC). For the forecast issued at T-60 hours, the cyclone landfall is located near 90°E, associated with the strong surge simulated over the surrounding area. As the atmospheric cyclone forecasts gradually converge towards the actual landfall location more than 100 km westward, so do our storm surge forecasts at T-36 hours and T-12 hours. The dependency of the timing of the landfall to the forecast range is notable here. At T-36 hours, the maximum water level pattern is similar to the hindcast shown in Figure 5. By T-12 hours, the magnitude of the forecast maximum water level corresponds well with the hindcast estimate.

## 6 Discussion

### 6.1 Coupled tide-surge-wave dynamics

The water level dynamics over the Bay of Bengal and particularly around the Bengal delta is complex, and the various components of water level have different relative contributions depending on the location. Due to tide-surge interaction, the maximum water level can be lower or higher depending on the tidal phase in a fully non-linear tide-surge coupled simulation, compared to its linear counterpart. During Amphan the tide-surge interaction was mostly negative, except within about 10 km of the

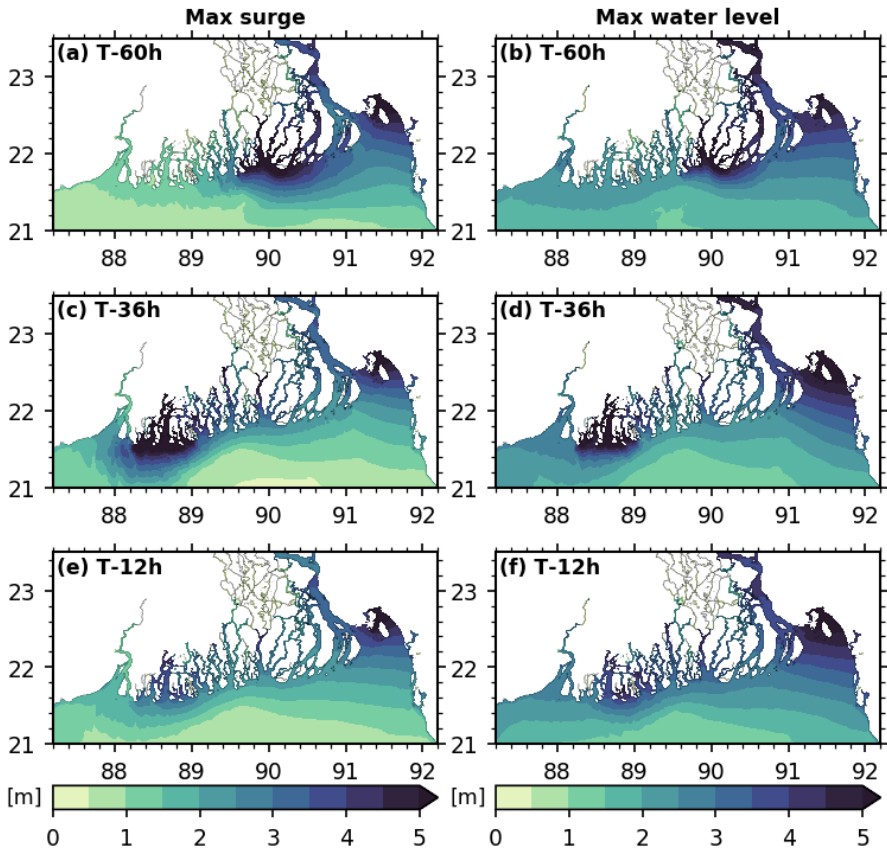

**Figure 7.** Maximum surge (a, c, e) and elevation (b, d, f) evolution for forecast initiated at (a-b) T-60 hours (2020-05-18 0000Z), (c-d) T-36 hours (2020-05-19 0000Z), (e-f) T-12 (2020-05-20 0000Z) hours before landfall.

landfall location. The positive interaction near the landfall location means that the coupled tide-surge estimate was higher than

the linearly added tide and surge. In this case, the typical magnitude of tide-surge interaction ranged from -30cm to +30cm. The contribution from the wave-induced setup along the shoreline increases the maximum water level estimation throughout the coast by about 10-15% during Amphan. Fully coupled tide-surge-wave models have been recommended for operational forecasting in the Bay of Bengal for years (Johns and Ali, 1980; Murty et al., 1986; As-Salek and Yasuda, 2001; Deb and Ferreira, 2016; Krien et al., 2017). Besides the development of the wave-ocean coupled hydrodynamic modelling tool itself,

the main challenge in implementing such a system sums up to the high computational requirements for the proposed modelling systems (Murty et al., 2014, 2016). Due to this constraints, operational systems are commonly run with either only surge, or only coupled tide-surge model, but without waves (Murty et al., 2017) and with a much coarser spatial resolution.


## 6.2 Performance of storm surge forecasting

Aside from the necessary inclusion of relevant physical processes – tide, surge, waves and their non-linear interactions – the
335 performance of storm surge forecasting depends on the wind and pressure forcings. The errors in the wind forcing and in the
atmospheric pressure forcing are typically the most important ones among the various sources of error in storm surge modelling
(Krien et al., 2017). In our forecast environment, the source of this error is two-fold – the analytical model used to synthesize
the fields from the storm parameters, and the numerical weather model used to forecast the storm wind and pressure fields to
derive the storm parameters themselves. There have been attempts to find a more accurate formulation from recent satellite
scatterometer missions (Krien et al., 2018). However, not one formulation works best in all radial distances from the storm
centre. Despite the well-known error associated with the parametric wind field, they are widely used in applications due to
their computational simplicity and lightweight data requirements (Lin and Chavas, 2012). The best way to avoid the error from
the analytical wind field might be not using these formulations and rely on the full-fledged atmospheric forecasts. However,
atmospheric models are costly in terms of computation. It is particularly true for cyclonic storms where high spatial resolution
(typically kilometric) is required (Tallapragada et al., 2014).

From the validation of water level at a few tide gauge locations for our hindcast simulation, it appears that the modelling
framework proposed here could capture the maximum surge level successfully throughout the coast. Our tide-gauge sites were
limited to the eastern side of Amphan landfall location and were operational throughout the life cycle of the cyclone. However,
it was not possible to confirm the ability of the model to reproduce the water level on the western side of the landfall location
due to data unavailability. The proper reproduction of the maximum water level by the model gives strong confidence in the
formulation as well as in the coupling strategy of our modelling framework.

## 6.3 Inundation

One of the potential, but challenging, outcomes of storm surge forecasting is the prediction of inundation. Particularly in the
northern Bay of Bengal, the existing modelling studies do not generally consider the inundation process (Dube et al., 2004;
Murty et al., 2014). Some modelling systems take advantage of simplified inundation modelling schemes to tackle this problem
(Lewis et al., 2013). While the recent improvements in models bathymetry and the detailed accounting of dikes and coastal
defences improved the overall modelling of the inundation, the associated error remains large (Krien et al., 2017). As an update
to Krien et al. (2017), in our model we have incorporated a novel dike heights dataset, bearing varying dike crests heights, for
the numerous polders scattered around the Bengal delta. The assessment of the impact of the updated embankment heights is,
however, hard to quantify and validate.

In the absence of any operational network of inundation monitoring, to understand and better characterize the inundation
dynamics, we systematically skimmed the Bengal delta local newspapers on the day of Amphan landfall and on the few sub-
sequent days, to achieve an as-comprehensive-as-possible mapping of the inundation extent observed in situ. We digitized the
reported flooded locations through © Google Earth and categorized the inundation mechanism. We digitized 88 such loca-
365 tions over the Bangladeshi part of Bengal delta, as shown in Figure 8 (Khan, 2020). Over the Indian side of the delta, the





news reporting of inland flooding from dike breaching was not accurate enough for us to be able to geotag the locations. We overlaid the inundated locations over the inundation predicted by the model on a false-colour image derived from Sentinel-2 satellite using © Google Earth Engine (https://earthengine.google.com). Three categories of inundation mechanism are typically observed – inundation by breaching of the dikes (labelled as "Embankment breach"), inundation of unprotected low

land by increased water level ("Unembanked lowland"), and flooding by overflowing of the dikes ("Embankment overflow"). From the reported news, it is clear that the major part of the inundation results from the breaching of dikes. We also note two breaching instances in the south-eastern corner of our domain, very far from the cyclone landfall. On the other hand, the flooding from merely increased water level in non-diked areas is mostly concentrated along the Meghna estuary. Along the estuary, designed embankments probably do not exist to protect many populated low-lying char areas (land/island formed though

accretion). We could find only one report of the occurrence of overflowing of an embankment, in Maheshwaripur of Koyra upazila (89.30°E, 22.45°N). Around (90.1°E-90.5°E, 22.4°N-22.6°N), in Barishal district, our model predicted an extended inundated area, which is known not to be polderized. While we could find some reports of flooding around that region from our newspaper survey, the widespread inundation predicted by our model in this region probably did not occur in reality due to the presence of levees and city protection embankments. Indeed, this kind of small-scale geographic information, to the best of our

knowledge, is not systematically available publicly and thus could not be incorporated in our model topography, despite being of utmost importance for localized inundation forecast. The forecast flooded vs non-flooded areas show a wealth of spatial scales, demonstrating that our unstructured modelling framework has the intrinsic capability to model the small-scale hydrodynamic gradients. In our modelled forecasts, the contrast between the well-predicted water level temporal evolution, and the comparatively limited predictive skills of inundated locations points out the necessity of accurate knowledge and incorporation

of reliable topographic and embankment information, at the local scale. Our modelling system has the ability and appropriate resolution to ingest such topographic information efficiently due to its unstructured nature.

The coastal polder management has long been a boon and a bane (Warner et al., 2018). On one side, the polders have protected the fertile areas from daily tidal flooding of brackish waters, which favoured agricultural production by reducing the soil salinity (Nowreen et al., 2013). On the other hand, these embankments have restricted freshwater and sediment supply,

causing significant subsidence inside the embanked areas due to sediment compaction (Auerbach et al., 2015). Over time, the land use inside these polders has also changed from agriculture to aquaculture (namely shrimp and crab farming). Although the embankments were not built to protect from the cyclonic surges, in many instances they functioned as such. The presence of these protective structures also contributed to a false sense of security in the residents living inside (Paul and Dutt, 2010). Embankments have become part and parcel of survival in this low-lying region. It has now become essential to monitor the

condition and topography of the embankments for efficient management and maintenance. From a forecasting point of view, embankment breaching is extremely hard to model in the state-of-the-art hydrodynamic modelling frameworks. However, consistent periodic monitoring of the embankment conditions can improve the forecast by providing a more objective view of the associated inundation hazard.


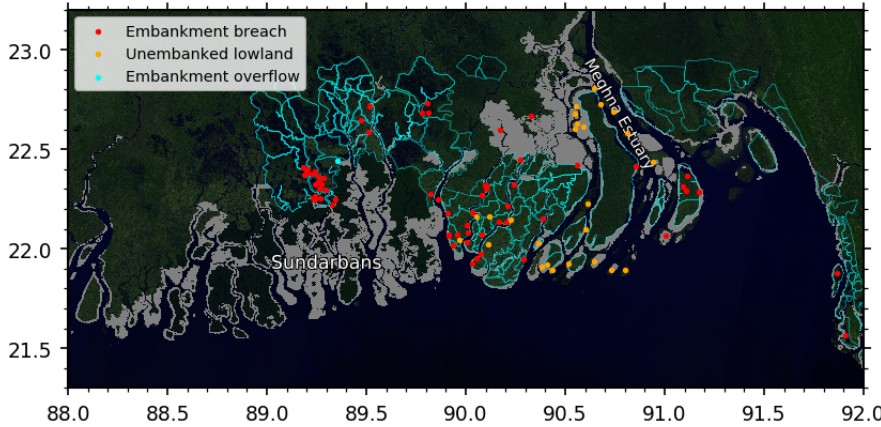

**Figure 8.** Digitized location of inundation resulting from embankment breaching (red), flooding of low-lying unembanked area (orange), and embankment overflow (cyan) overlaid on a false-colour image composite derived from B12, B11, and B4 channels of Sentinel-2 during April 2020. The grey patches are the inundated regions predicted by the model, as shown in the hindcast experiment. The polders are shown with cyan outlines.

## 6.4 Prospects in operational forecasting

Throughout our forecast and hindcast experiments, we have relied on freely available forecast products as the forcing for our storm surge model. These forecasts data are publicly available within 3 hours of their initialization time through online portals. The primary input to generate the forcing is the forecast storm track distributed as a text file, which is very lightweight even for a limited internet connection. We have implemented our storm surge model on a freely available open-source framework. The model code is already operationally used in regional forecasting systems and services (Fortunato et al., 2017; Oliveira et al.,

2020).

As we have demonstrated, the modelling setup presented here is tractable in an operational, real-time forecasting scenario. Thanks to advanced and efficient numerics, one can deploy such a system in a consumer-grade computing environment. The computing setup we have deliberately used (see Table 1) essentially amounts to a high-end x86-64 desktop personal computer (e.g. Intel® Core™i9-10900 or similar). This computing requirement takes only a portion of the available computing capacity

of an institute such as the BMD (15x4 cores@2.8GHz) (Roy et al., 2015). Furthermore, the model is easily deployable in cloud computing infrastructure (Oliveira et al., 2020), which could provide additional reliability and cost-savings for an event-driven operational forecasting system.

## 7 Conclusions

In this study, we present a retrospective evaluation of storm surge prediction during super-cyclone Amphan using an efficient,

state-of-the-art coupled hydrodynamic-wave numerical model. During cyclone Amphan, the predicted maximum water level




ranges from 2 to 5 m along the coastline. Comparison with in-situ measurements revealed that the water level could be modelled with high accuracy on condition that all the relevant mechanisms are considered. Notably, we demonstrated the necessity of considering a coupled hydrodynamic tide-surge-wave modelling framework for the head Bay of Bengal, for effective forecasting. For Amphan, the contribution from tide-surge interaction, and wave-setup typically amounts to 10% and 10-15% of the total water level, respectively.

From our proactive forecast initiative during cyclone Amphan, we showed that with publicly available storm forecast products and easily accessible computing resources, it is feasible to forecast the evolution of water level throughout the vast coastline of the Bengal delta in real-time. During Amphan, a sufficiently skilful storm surge forecast was achieved as early as 36-48 hours before landfall. The forecast water level with 36-hour lead time seemed quite similar to our best (hindcast) simulation, in terms of maximum water level as well as the spatial pattern.

From a secondary post-disaster news-survey, we have identified the limiting factors for location-specific inundation forecasts. The main limiting factor is due to limited and outdated topographic information of the existing coastal defences. Also, dike-breaching, which was the prevalent process of inland inundation during Amphan, is not explicitly modelled in our hydrodynamic framework. These two factors call for routine monitoring of embankment topography and condition.

Cyclone and storm surge warning has always been a communication and trust issue in the Bay of Bengal (Paul and Dutt, 2010; Roy et al., 2015). It is thus necessary to communicate well-grounded storm and surge forecast to the community for coordinated and informed decision-making during a storm surge (Morss et al., 2018). The forecasting system we implemented and assessed in the present case study provides the proof-of-feasibility and opens short-term operational prospects to fill a gap in the existing disaster management tools in this part of the world.

*Code and data availability.* The instructions to download and install the model used in this study can be accessed freely at https://github.com/schism-dev/schism. The sources of the data used in this study have been described in the article. The data should be requested through the mentioned institutions or downloaded from the provided websites. The geolocation of inundation mapping from newspaper can be found at https://dx.doi.org/10.5281/zenodo.4086102.

## Appendix A

### A1  Tide model validation

Our tidal model is validated at 7 tide gauge locations, around the Bengal delta. We have used complex error as the performance indicator (Mayet et al., 2013). The harmonic analysis is done using the Tidal toolbox developed at LEGOS (Allain, 2016). The modulus of the complex difference defines the complex error for a tidal constituent.

$$|\Delta z| = |A_m e^{i\phi_m} - A_o e^{i\phi_o}| \tag{A1}$$





Where $A$ and $\phi$ are the amplitude and phase (in radians) respectively, of the tidal harmonics. The subscript denotes the model ($m$) and observation ($o$). The total error of all the constituent at one location is calculated as the squared root of half of the squared sum.

$$\sigma_s = \sqrt{\frac{1}{2}\sum_N |\Delta z|^2} \qquad\qquad (A2)$$

    Along the coast of Bengal delta, only four of the constituents - M2, S2, K1, and O1 are found to contribute significantly to
the tidal energy (Sindhu and Unnikrishnan, 2013). As in many cases, information for other tidal harmonics is not available, only these four constituents are considered for calculating the total complex error at a location.

    A comparison of the complex error between the global models and the model presented here is shown in Table A1. Amplitudes (A) and errors are in centimeter, phase ($\phi$) is in degrees. Hooghly River, Diamond Harbour, Garden Reach and Chandpur are not represented in global tidal models (FES, GOT, and TPXO) due to their location in far upstream.

**A2   Forcing data source**

The source of the forcing data used in this study for hindcast and forecast is listed in Table A2

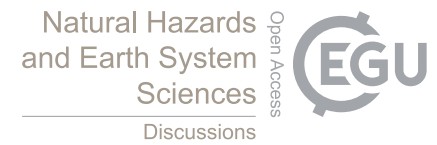
**Table A1.** Performance of tidal model at tide-gauge locations.

| Station | | Observation | | FES2012-Hydro | | | FES2012 | | | GOT4.8 | | | TPXO7.2 | | | Krien et al. (2016) | | | This Model | | |
|---|---|---|---|---|---|---|---|---|---|---|---|---|---|---|---|---|---|---|---|---|---|
| | | $A_0$ | $\phi_0$ | $A_m$ | $\phi_m$ | Error | $A_m$ | $\phi_m$ | Error | $A_m$ | $\phi_m$ | Error | $A_m$ | $\phi_m$ | Error | $A_m$ | $\phi_m$ | Error | $A_m$ | $\phi_m$ | Error |
| Sagar Roads (88.0300°E, 21.6500°N) | M2 | 140 | 116 | 142 | 99 | 42 | 137 | 104 | 29 | 113 | 113 | 27 | 132 | 104 | 28 | 143 | 116 | 3 | 144.5 | 114.9 | 5.3 |
| | S2 | 66 | 150 | 73 | 141 | 13 | 62 | 141 | 11 | 40 | 145 | 40 | 48 | 126 | 29 | 62 | 155 | 7 | 62.4 | 153.3 | 5.2 |
| | K1 | 15 | 262 | 17 | 256 | 2 | 16 | 253 | 3 | 14 | 277 | 14 | 14 | 258 | 1 | 17 | 265 | 2 | 15.6 | 265.4 | 1.1 |
| | O1 | 5 | 250 | 6 | 251 | 1 | 6 | 243 | 1 | 5 | 270 | 2 | 5 | 252 | 0.4 | 6 | 248 | 1 | 5.7 | 251.6 | 0.8 |
| | $\sigma_s$ | | | | | 31 | | | 22 | | | 27 | | | 29 | | | 6 | | | 5.3 |
| Diamond Harbour (88.1733°E, 22.1928°N) | M2 | 157 | 168 | | | | | | | | | | | | | 166 | 161 | 21 | 142.3 | 165.6 | 15.9 |
| | S2 | 68 | 210 | | | | | | | | | | | | | 68 | 207 | 4 | 57.6 | 208.6 | 10.4 |
| | K1 | 15 | 285 | | | | | | | | | | | | | 16 | 284 | 1 | 13.2 | 286.3 | 1.8 |
| | O1 | 7 | 258 | | | | | | | | | | | | | 5 | 253 | 2 | 5.4 | 257.7 | 1.6 |
| | $\sigma_s$ | | | | | | | | | | | | | | | | | 15 | | | 13.6 |
| Hiron Point (89.4780°E, 21.8169°N) | M2 | 81 | 127 | 86 | 88 | 56 | 87 | 91 | 52 | 80 | 88 | 53 | 104 | 110 | 35 | 81 | 115 | 17 | 99.9 | 115.0 | 26.7 |
| | S2 | 34 | 159 | 45 | 121 | 28 | 40 | 122 | 24 | 37 | 118 | 25 | 37 | 136 | 14 | 35 | 148 | 7 | 41.6 | 150.5 | 9.3 |
| | K1 | 13 | 268 | 15 | 250 | 5 | 16 | 252 | 5 | 14 | 248 | 5 | 14 | 261 | 2 | 15 | 265 | 2 | 15.0 | 265.7 | 1.7 |
| | O1 | 5 | 258 | 6 | 244 | 2 | 6 | 238 | 2 | 5 | 244 | 1 | 5 | 256 | 0.3 | 6 | 245 | 1 | 5.7 | 255.0 | 0.7 |
| | $\sigma_s$ | | | | | 44 | | | 40 | | | 42 | | | 27 | | | 13 | | | 20.0 |
| Dhulasar (90.2700°E, 21.8500°N) | M2 | 73 | 158 | 58 | 114 | 52 | 80 | 117 | 53 | 79 | 117 | 54 | 86 | 121 | 51 | 51 | 156 | 22 | 67.6 | 143.3 | 18.8 |
| | S2 | 35 | 193 | 39 | 141 | 33 | 39 | 142 | 32 | 39 | 146 | 29 | 35 | 135 | 34 | 20 | 194 | 15 | 28.5 | 179.6 | 9.8 |
| | K1 | 13 | 286 | 15 | 262 | 6 | 16 | 256 | 8 | 15 | 260 | 6 | 15 | 255 | 8 | 12 | 297 | 3 | 13.3 | 287.8 | 0.5 |
| | O1 | 4 | 278 | 6 | 256 | 3 | 6 | 243 | 3 | 6 | 256 | 3 | 6 | 250 | 3 | 5 | 280 | 1 | 5.6 | 273.8 | 1.6 |
| | $\sigma_s$ | | | | | 44 | | | 44 | | | 44 | | | 44 | | | 19 | | | 15.0 |
| Charchanga (91.0500°E, 22.2188°N) | M2 | 96 | 234 | 110 | 202 | 57 | 115 | 208 | 50 | 97 | 204 | 49 | 84 | 154 | 103 | 67 | 208 | 46 | 95.8 | 216.9 | 28.5 |
| | S2 | 37.5 | 265 | 38 | 238 | 18 | 30 | 243 | 15 | 34 | 234 | 19 | 36 | 186 | 47 | 27 | 241 | 17 | 36.6 | 250.3 | 9.5 |
| | K1 | 13 | 304 | 17 | 298 | 4 | 16 | 300 | 4 | 7 | 314 | 6 | 16 | 272 | 8 | 14 | 309 | 2 | 16.8 | 308.7 | 4.0 |
| | O1 | 8 | 285 | 7 | 289 | 1 | 6 | 284 | 2 | 4 | 303 | 4 | 6 | 267 | 3 | 8 | 289 | 0 | 8.1 | 293.1 | 1.1 |
| | $\sigma_s$ | | | | | 43 | | | 37 | | | 37 | | | 80 | | | 35 | | | 21.5 |
| Chittagong (91.8274°E, 22.2434°N) | M2 | 173 | 196 | 118 | 193 | 56 | 126 | 200 | 49 | 120 | 192 | 54 | 89 | 153 | 123 | 156 | 198 | 18 | 149.2 | 194.8 | 24.1 |
| | S2 | 64 | 229 | 41 | 230 | 23 | 33 | 236 | 31 | 43 | 227 | 21 | 40 | 160 | 62 | 58 | 235 | 9 | 55.0 | 225.8 | 9.6 |
| | K1 | 19 | 278 | 17 | 294 | 6 | 17 | 295 | 6 | 9 | 300 | 11 | 16 | 258 | 7 | 20 | 289 | 4 | 19.1 | 284.9 | 2.3 |
| | O1 | 8 | 263 | 7 | 285 | 3 | 6 | 280 | 3 | 4 | 289 | 5 | 6 | 252 | 2 | 8 | 269 | 1 | 7.9 | 267.3 | 0.6 |
| | $\sigma_s$ | | | | | 43 | | | 41 | | | 42 | | | 98 | | | 14 | | | 18.4 |
| Chandpur (90.6385°E, 23.2344°N) | M2 | 29.7 | 31.4 | | | | | | | | | | | | | | | | 33.6 | 333.7 | 30.7 |
| | S2 | 10.5 | 62.3 | | | | | | | | | | | | | | | | 11.2 | 6.3 | 10.2 |
| | K1 | 5.6 | 18.6 | | | | | | | | | | | | | | | | 5.4 | 21.9 | 0.7 |
| | O1 | 3.4 | 12.9 | | | | | | | | | | | | | | | | 3.6 | 357.4 | 1 |
| | $\sigma_s$ | | | | | | | | | | | | | | | | | | | | 22.9 |




**Table A2.** Data and sources for the model for forcings

| Name | Datatype | Source |
| --- | --- | --- |
| JTWC advisory | Text | https://www.metoc.navy.mil/jtwc/jtwc.html |
| HWRF forecast | Text | https://nomads.ncep.noaa.gov:9090/dods/gfs_0p25 |
| GFS | 4D array/DAP | https://nomads.ncep.noaa.gov/dods/gfs_0p25_1hr |
| FNL | netCDF | https://rda.ucar.edu/datasets/ds083.3/ |



*Author contributions.* JK and FD formulated the study. JK did the numerical simulations, analysis, and wrote the first draft. XB computed the simulation for WW3 boundary used in hindcast. All co-authors contributed to this study through multiple discussions. MP provided help with the WWM formulation. SH disseminated the forecasts to local government of Bangladesh, and ASI to the scientific community during cyclone Amphan.

*Competing interests.* The authors declare that they have no conflict of interest.

*Acknowledgements.* We acknowledge financial support from CNES (through the TOSCA project BANDINO) and Embassy of France in Bangladesh. This work was supported by the French research agency (Agence Nationale de la Recherche; ANR) under the DELTA project (ANR-17-CE03-0001). We are thankful to LIENSs Laboratory (University of La Rochelle, France) for hosting JK and LT during this study.





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
