# Peer review of "Towards an efficient storm surge and inundation forecasting system over the Bengal delta: Chasing the super-cyclone Amphan"

_Natural Hazards and Earth System Sciences, 2020_

## Referee Comment (RC1) · Anonymous Referee #1 · 19 Nov 2020

Major comments: Page 1: Line 18: It is mentioned as 'Amphan struck the coasts of Bangladesh and India'. But from figure 1, it is clear that the track has crossed the Indian coast and passes through the Bangladesh mainland, but not actually crossed the Bangladesh coast. Modify the sentence accordingly. Page 3: Line 57: 'reduced-physics modeling', what it means? It seems Murty et al. 2017 used the ADCIRC+SWAN model to investigate the wave, surge interaction in the shallow waters. Page 4: Lines 96-98: Explaining the tide surge interaction. I suggest citing the article Srinivasa Kumar et al. 2015 entitled 'Modeling Storm Surge and it's Associated Inland Inundation Extent Due to Very Severe Cyclonic Storm Phailin' here. Their work was clearly investigated how the phase of the tide alters the surge height and inundation

extent. I feel this citation suits here. However, this is up to the author's choice. Page 7: Line 164: What is the resolution of the mentioned bathymetry? Whether the 77,000 points are really sufficient to cover the entire Bengal and Bangladesh coasts especially while computing the inundation extent? Page 8: Line 173: Is this the whole Bay of Bengal or part of it? Because the latitude extents given in the brackets don't cover the entire BoB. Please check. Page 8: Line 175: 250m resolution near the coast is acceptable for the surge computations. However, is it sufficient for the inland inundation computations? Page 8: Line 176: Mentioned here that the model domain and mesh are shown in figure 9, but no figure in the draft shows the mesh and domain. It seems the figure is missed. The domain and mesh figure are important for the readers and hence it should be provided. Page 8: Line 178: Mentioned here as 'wave model, is coupled online with SCHISM'. What does it mean? Whether the wave model also uses the same unstructured mesh or it uses the structured mesh? Whether both the models are running at a time (i.e., in parallel)? or running the wave model and then transfer the wave boundary condition to the surge model? These points are to be briefly explained. The given citation Roland et al., 2012 can be used for the complete details. Page 8: Line 200: Given that the blended wind field is used. What is the horizontal resolution of the wind and pressure fields? It seems the tide is also included in the computations. What is the spinup time used in the study to get the actual tide levels at the coast? What is the source of the buoy data used in the study? Figure 4: There is a clear mismatch between observation and modeled total water level at the given locations especially at Angtihara and Tajumuddin. This might be due to the lack of spinup for tide simulation. The reason given here is 'The local bathymetric error and friction parameterization might be the source of the discrepancy'. But the same model used by Krien et al. while using the digitized sounding points has computed the better tide amplitudes. Please check the spinup time. As mentioned if Angtihara is located in a data-scarce location inside Sundarbans mangrove forest, remove the plot. Provide the water level - tide (surge residual) plots (time series) too, to support the statement in the line numbers 246-247, page 11. Inundation section: The methodology is to be

clearer. Though Lewis et al. is cited for the details, a brief description is required here. Whether the model mesh extends on to the land or not? if so up to what extent (i.e., up to which topography contour)? or whether the water level values at the coast are used and extrapolated the inundation extents?

Please also note the supplement to this comment:
https://nhess.copernicus.org/preprints/nhess-2020-340/nhess-2020-340-RC1-supplement.pdf

---

## Referee Comment (RC2) · Anonymous Referee #2 · 7 Dec 2020

This paper presents an interesting strategy to address surge and inundation in the bay of Bengal, based on publicly available model and forcings. The quality of the results is much larger than the previous efforts and a detailed analysis on process dominance is presented. The proposed strategy, although apparently customized to this site, is interesting and worth publishing.

Below I have outline major and minor issues to be solved before approval, some of them requiring new runs. Furthermore, the paper is extremely long (sometimes it looks more a report than a scientific paper) and lacks now and then a clear presentation direction. The paper should thus be reviewer for conciseness and easy of reading.

I consider the paper to be accepted with major reviews.

Abstract

- "Despite recent advancements, the complex morphology and hydrodynamics of this large delta and the associated modelling computational costs impede the storm surge forecasting in this highly vulnerable region." – Nowadays, super computers are available to perform forecasts of much larger and more complex domains, integrated with atmospheric models. The author should review this sentence maybe focusing on the quality of the forecasts and the necessary grid refinements and process-knowledge for high quality results.
- "This article shows the proof of the maturity of our framework for operational implementation, which can particularly improve the quality of localized forecast for effective decision-making" – Is the framework generic or only applicable to the bay of Bengal? The authors should clarify this issue at the abstract.

Line 36

- "global weather and forecasting system" – a word is missing of the "the" word needs to be removed

Lines 44-45

- "Nowadays, operational surge forecasting systems typically run on high-performance computing systems, either on a scheduled basis or triggered on-demand during an event (Khalid and Ferreira, 2020)" – The authors should include other references of such system, either applicable at a specific site or of generic application.

Lines 50-52

- "Storm surge forecasts have shown their potential to better target the evacuation decision, to optimize early-engineering preparations, and to improve the efficiency of the allocation of the resources (Glahn et al., 2009; Lazo and Waldman, 2011)." – again an updated and more comprehensive review is missing, along with the identification of what are the major challenges in developing and keeping in operational mode this type of systems.

Lines 58-60

- "In the past decade, unstructured-grid modelling systems are getting more and more popular due to their efficiency in resolving the topographic features and their reduced computational cost compared to structured-grid equivalents (Ji et al., 2009; Lane et al., 60 2009; Melton et al., 2009)." – all these references are not from the last decade. Part of them do not address operational forecast systems. There are several examples of unstructured grid forecast system in operation, some recent some in operation for over a decade. The authors should review carefully the state-of-the-art and improve the current text.

Line 100

- "Due to this interaction, the highest surge is obtained for a storm making landfall around 2 hours before the high tide." – a detailed explanation (or references explaining it) is needed. Is it associated with the specifics of the geometric/bathymetric characteristics of the bay or generic? Is it tidal amplitude dependent?

Line 165-169

- "Our bathymetric dataset is a blend of two digitized sounding datasets in the nearshore zone – one from navigational charts produced by Bangladesh Navy, and another being a bathymetry of the Hooghly estuary provided by IWAI (Inland Waterways Authority of India)"- are there any common areas between the two sources ? if yes, what was the combination procedure? If not, substitute "blend" by "combination". How old is the data?
- "The river bathymetry is composed of a set of cross-sections obtained from the Bangladesh Water Development Board (BWDB)." – what is the spacing between profiles? How old is the data?

Lines 194-195

- "At each of the upstream 195 river open boundaries of Ganges, Brahmaputra, Hooghly, and Karnaphuli, we implemented a discharge boundary condition" – what is the source of the discharge values?

Fig 4 – the map is unreadable. Place it at a larger scale.

Lines 200 to 209

this paper aims at evaluating a procedure (framework is not adequate in this context) to forecast storm surges and evaluates the procedure using a past event. However when running an operational model, reanalysis are not available. Therefore the quality of the model should be evaluated with a past event (so data is available) but for a run under operational forecast conditions. The analysis of fig 4 should therefore by re-done under these conditions.

Lines 213 and following: why is the comparison limited to the storm path? The data is available in the whole domain.

Lines 274 and following: this text belongs in the introduction. Remove.

Lines 282-284: "We communicated the results to Bangladesh local government authority through personal communications, as well as to the scientific community through social media." – this sentence is off context and has no scientific link with the remaining of the chapter. It should be moved to other parts of the paper (introduction?)

Fig 6 only deals with time. It should be improved with a plot on space definition of forcings. The use of the "blend" should be reviewed.

Page 13 – the proposed methodology seems too linked with the specific physics of the site and of this particular event. Small variations of the coupling should be tested and compared with data.

Lines 315 and following: errors are necessary for the forecast runs. The discussion is very weakly supported without them.

Lines 327 until the end of page 15 belong in the introduction as motivation for this study.

Lines 344: "The best way to avoid the error from the analytical wind field might be not using these formulations and rely on the full-fledged atmospheric forecasts". In spite of the limitations of existing atmospheric forecasts, this sentence and the next ones would be better supported with a simulation just based on the available forecasts. I suggest the author to repeat the simulation without the analytical model and evaluate the differences.

Fig 8 – why are the inundation patches for the hindcast experiment and not for the forecast runs? I suggest that those are included too, with another figure.

Lines 400-401 – refer to table A2. In table A2, correct the 2nd link as it is not a public link.

---

## Author Comment (AC1) · 2 Feb 2021

**Replies to comments of Reviewer #1**

**Major comments:**

Page 1: Line 18: It is mentioned as 'Amphan struck the coasts of Bangladesh and India'. But from figure 1, it is clear that the track has crossed the Indian coast and passes through the Bangladesh mainland, but not actually crossed the Bangladesh coast. Modify the sentence accordingly.

Reply: We acknowledge that the center of the cyclone Amphan made landfall in West Bengal, India. However, the overall impact of the storm, be it wind (34 knots radial band in Figure 1) or associated surge (Figure 5), largely exceeded the borders of West Bengal, and spanned the shorelines of both India and Bangladesh. Thus, we tried to avoid the common wording about "landfalling" and used "struck" to indicate that both India and Bangladesh were impacted by Amphan. We propose to revise the line as follows:

"… Amphan made landfall in West Bengal (India), with a sustained wind speed of 112 km per hour and gusts of 190 km per hour, causing massive damage in India and Bangladesh and claiming at least 116 lives."

Page 3: Line 57: 'reduced-physics modeling', what it means? It seems Murty et al. 2017 used the ADCIRC+SWAN model to investigate the wave, surge interaction in the shallow waters.

Reply: By "reduced-physics modelling" we meant a storm surge model without coupling with a wave model. For clarity, we propose to replace "reduced-physics modelling" by "modelling without waves coupling". From our reading, Murty et al. (2017) did not use any storm-surge - waves coupled model, but only a storm surge model (ADCIRC) for real-time forecasting (which is the concern of this particular statement). On a related topic, we acknowledge that Murty et al. (2016) did use a storm-surge - waves coupled model (ADCIRC+SWAN) to investigate storm surges and wave setup for cyclone Hudhud. It is noteworthy that both these papers are cited in the manuscript.

Page 4: Lines 96-98: Explaining the tide surge interaction. I suggest citing the article Srinivasa Kumar et al. 2015 entitled 'Modeling Storm Surge and it's Associated Inland Inundation Extent Due to Very Severe Cyclonic Storm Phailin' here. Their work was clearly investigated how the phase of the tide alters the surge height and inundation extent. I feel this citation suits here. However, this is up to the author's choice.

Reply: Thank you for the suggested reference. We propose to cite it accordingly in the corresponding part of the manuscript at L96-98

Page 7: Line 164: What is the resolution of the mentioned bathymetry? Whether the 77,000 points are really sufficient to cover the entire Bengal and Bangladesh coasts especially while computing the inundation extent?

Reply: As we have mentioned in the manuscript (L164), the current bathymetry is an updated version of the bathymetry produced by Krien et al. (2016).

[Figure]

Fig C1. (a) Navigational charts of NHO (India) digitized in this study (white rectangles) as well as maps given by Bangladesh Navy and Mongla Port Authorities (BN and MPA respectively). (b) Display of digitized data (in white) and data from bathymetric surveys in rivers (in red). (c) Zoom of digitized data in the Hooghly river. (Adapted from Krien et al., 2016).

This dataset is complemented by additional 77k digitized points, which is shown in Fig C2. The resolution of the chart points ranges from 200 m to 5 km, with a prevalence of 300-500 m as the most common spacing.

[Figure]

Fig C2. Digitized sounding points from Bangladesh Navy charts (in yellow), the red boxes show the individual chart outlines.

These digitized points do not cover the inland area, which is separately covered by a high-resolution (50 m) inland topography dataset over the south-central part of the delta on Bangladesh side (as explained in L169), and rest is complemented by SRTM dataset (refer to L171). The extent of the CEGIS data over inland is shown in Fig C3.

[Figure]

Fig C3. Extent of the 50 m resolution inland topography dataset (in red).

We propose to include Fig. C2 and Fig. C3 as Supplementary materials.

Page 8: Line 173: Is this the whole Bay of Bengal or part of it? Because the latitude extents given in the brackets don't cover the entire BoB. Please check.

Reply: As stated in the manuscript, our domain covers the Bay of Bengal to the North of 11°N. We agree that traditionally, the commonly accepted geographical limit of the Bay of Bengal is rather around 6°N to 8°N. To remove this ambiguity, we propose to reword "whole Bay of Bengal" to "northern Bay of Bengal".

Page 8: Line 175: 250m resolution near the coast is acceptable for the surge computations. However, is it sufficient for the inland inundation computations?

Reply: Given the current level of detail available for the inland topography, we firmly think that 250 m is a well-suited resolution for inland inundation computation. This resolution is not dictated by the scale of the hydrodynamic features, but rather by the resolution of the topographic databases available over our domain. Indeed, the currently available topographic datasets remain rather coarse, in that they do not resolve the sharp man-made physical features – e.g., embankments, roads etc. By combining separate datasets of embankment geometry and using the flexibility of the unstructured grid, we tried to capture the outlines and heights of the embankments in our model grid. As we showed in Figure 8, these embankments pose the zeroth-order control on the inland inundation, and yet we have limitations of the knowledge of the up-to-date embankment's heights.

However, at a much finer scale, one will think of implementing road-networks, associated water control structures (culverts, bridges etc.) once such high-resolution information becomes available.

Page 8: Line 176: Mentioned here that the model domain and mesh are shown in figure 9, but no figure in the draft shows the mesh and domain. It seems the figure is missed. The domain and mesh figure are important for the readers and hence it should be provided.

Reply: We are sorry for the misunderstanding here; we were referring to Figure 9 of Khan et al. (2019). Our sentence reads correctly in this regard. This being said, we agree with the reviewer that the inclusion of the figure will make it more convincing to the reader. We propose to include the following Figure C4 showing model domain, mesh, and boundary conditions as Figure 4 in the manuscript.

[Figure]

Figure C4. Computational domains and model mesh for SCHISM-WWMIII, as well as model boundary conditions. White arrows on the southern boundary show the forcing with the tidal solution provided by FES2012, and on the northern boundary shows the river discharges. For a hindcast experiment, wave spectra from WW3 are imposed on the southern boundary.

Page 8: Line 178: Mentioned here as 'wave model, is coupled online with SCHISM'. What does it mean? Whether the wave model also uses the same unstructured mesh or it uses the structured mesh? Whether both the models are running at a time (i.e., in parallel)? or running the wave model and then transfer the wave boundary condition to the surge model? These points are to be briefly explained. The given citation Roland et al., 2012 can be used for the complete details.

Reply: By 'coupled online' we meant that two models are fully coupled (and run as a unique executable, coupled at source code level without any external coupler). To avoid ambiguity, we propose to reword the segment as 'coupled at source code level'.

The wave model uses the same unstructured grid, as mentioned in the following line (L179).

The hydrodynamic core and the wave model run sequentially. To reflect this, we propose to update L183-184 'Water level, and current are exchanged among the two models every 30 minutes.' to 'Every 30 minutes of SCHISM runtime, water level and currents are passed to WWM for calculating the evolution of the wave fields. Calculated wave radiation stresses, total surface stress, and the wave orbital velocity are passed back to SCHISM before computing the next time step.'

Page 8: Line 200: Given that the blended wind field is used. What is the horizontal resolution of the wind and pressure fields?

Reply: The final resolution of the gridded wind and pressure field is 0.025° (roughly equivalent to 2.8km). We propose to add the following line at the end of L209 – 'The final resolution after merging the analytical wind field with the interpolated background GFS fields is 0.025°.'

Following another review comment, we have updated Figure 6 with a schematic of the spatial merging of wind and pressure fields from various sources. Considering these two revisions, we also propose to update L107-108 as 'The analytical and background wind fields were first temporally interpolated every 15 minutes and overlaid on the background GFS fields using a distance-varying weighting coefficient (from 3xRm to 10xRm, Rm is the radius of maximum wind) to ensure a smooth transition (Figure 6).'

It seems the tide is also included in the computations. What is the spinup time used in the study to get the actual tide levels at the coast?

Reply: Yes, we confirm that the tide is included in the computations, as stated in line 191-194. The spinup time is 2 days or longer in all our numerical simulations (either in hindcast mode, Section 4, or in forecast mode, Section 5). We propose to add the following sentence to make it clear (at L209):

"… Lin and Chavas (2012). For all storm surge simulations, a spinup time of 2 days is considered in this study."

What is the source of the buoy data used in the study?

Reply: buoy data used in this study is collected from INCOIS data portal (https://incois.gov.in/portal/datainfo/mb.jsp) during the time of the study which is mentioned in L222.

Figure 4: There is a clear mismatch between observation and modeled total water level at the given locations especially at Angtihara and Tajumuddin. This might be due to the lack of spinup for tide simulation. The reason given here is 'The local bathymetric error and friction parameterization might be the source of the discrepancy'. But the same model used by Krien et al. while using the digitized sounding points has computed the better tide amplitudes. Please check the spinup time. As mentioned if Angtihara is located in a data-scarce location inside Sundarbans mangrove forest, remove the plot.

Reply: The spinup time for the model (including the tides) is of 2 days, which we believe is much more than the typical time needed for a full tidal spin-up (inferior to 24 h over our domain in our model; not shown). We do not agree that Krien et al. (2016) obtained better tide amplitudes than ours. As seen in our Table A1, their model performance was generally similar to ours, and for several stations worse than ours. What is more, they did not assess the performance of their model in any of these two stations of Angtihara and Tajumuddin. However, we acknowledge that the plot of Angtihara time-series is perhaps misleading, and we agree to remove it.

Provide the water level - tide (surge residual) plots (time series) too, to support the statement in the line numbers 246-247, page 11.

Reply: Given the very limited length of observed tide gauge records that were available to us (13 days, including time period with surges), it is unfortunately not possible to operate a meaningful tidal analysis. Hence it is not feasible to compute any observed surge residual, although we agree it would have improved the clarity of this statement.

**Inundation section:** The methodology is to be clearer. Though Lewis et al. is cited for the details, a brief description is required here. Whether the model mesh extends on to the land or not? if so up to what extent (i.e., up to which topography contour)? or whether the water level values at the coast are used and extrapolated the inundation extents?

Reply: The model extent of the model mesh is now shown in a new Figure (Figure C4). The model mesh does extend over land, up to beyond the topography contour of 5 m above MSL.

Regarding the calculation methodology, it seems there is a misunderstanding as we did not cite Lewis et al. 2013 "for details" relevant to our own modeling framework. Indeed, unlike Lewis et al. 2013, inundation is modelled seamlessly in SCHISM, considering the same hydrodynamics as estuaries or ocean. To clarify the methodology and avoid misunderstanding we propose to add the following line at L356 –

'… (Lewis et al. 2013). In our modelling framework, the inland inundation is calculated seamlessly by SCHISM, solving the same hydrodynamics over the model domain, thanks to its wetting-drying algorithm. While the recent improvement…'

---

## Author Comment (AC2) · 2 Feb 2021

A point-by-point reply to the reviewer's comments is provided in the supplement pdf.

Please also note the supplement to this comment:
https://nhess.copernicus.org/preprints/nhess-2020-340/nhess-2020-340-AC2-supplement.pdf

---

## Author Response (AR1)

**Replies to comments of Reviewer #1**

**Major comments:**

Page 1: Line 18: It is mentioned as 'Amphan struck the coasts of Bangladesh and India'. But from figure 1, it is clear that the track has crossed the Indian coast and passes through the Bangladesh mainland, but not actually crossed the Bangladesh coast. Modify the sentence accordingly.

Reply: We acknowledge that the center of the cyclone Amphan made landfall in West Bengal, India. However, the overall impact of the storm, be it wind (34 knots radial band in Figure 1 of the revised manuscript) or associated surge (Figure 6 of revised manuscript), largely exceeded the borders of West Bengal, and spanned the shorelines of both India and Bangladesh. Thus, we tried to avoid the common wording about "landfalling" and used "struck" to indicate that both India and Bangladesh were impacted by Amphan. We have revised the lines as follows (L18-20 of revised manuscript):

"… Amphan made landfall in West Bengal (India), with a sustained wind speed of 112 km per hour and gusts of 190 km per hour, causing massive damage in India and Bangladesh and claiming at least 116 lives."

Page 3: Line 57: 'reduced-physics modeling', what it means? It seems Murty et al. 2017 used the ADCIRC+SWAN model to investigate the wave, surge interaction in the shallow waters.

Reply: By "reduced-physics modelling" we meant a storm surge model without coupling with a wave model. For clarity, we have replaced "reduced-physics modelling" by "modelling without waves coupling".

From our reading, Murty et al. (2017) did not use any storm-surge - waves coupled model, but only a storm surge model (ADCIRC) for real-time forecasting (which is the concern of this particular statement). On a related topic, we acknowledge that Murty et al. (2016) did use a storm-surge - waves coupled model (ADCIRC+SWAN) to investigate storm surges and wave setup for cyclone Hudhud. It is noteworthy that both these papers are cited in the manuscript (e.g., L81,113 of the revised manuscript).

Page 4: Lines 96-98: Explaining the tide surge interaction. I suggest citing the article Srinivasa Kumar et al. 2015 entitled 'Modeling Storm Surge and it's Associated Inland Inundation Extent Due to Very Severe Cyclonic Storm Phailin' here. Their work was clearly investigated how the phase of the tide alters the surge height and inundation extent. I feel this citation suits here. However, this is up to the author's choice.

Reply: Thank you for the suggested reference. We cited it accordingly in the corresponding part of the manuscript (L113 of the revised manuscript).

 What is the resolution of the mentioned bathymetry? Whether the 77,000 points are really sufficient to cover the entire Bengal and Bangladesh coasts especially while computing the inundation extent?

Reply: As we have mentioned in the manuscript (L164 in the old, L175 in the revised manuscript), the current bathymetry is an revised and updated version of the bathymetry assembled by Krien et al. (2016). Their coverage is shown in Fig C1.

[Figure]

Fig C1. (a) Navigational charts of NHO (India) digitized in Krien et al., 2016 (white rectangles) as well as maps given by Bangladesh Navy and Mongla Port Authorities (BN and MPA respectively). (b) Display of digitized data (in white) and data from bathymetric surveys in rivers (in red). (c) Zoom of digitized data in the Hooghly river. (Adapted from Krien et al., 2016).

This dataset is complemented by additional 77k digitized points, which is shown in Fig C2. The resolution of the chart points ranges from 200 m to 5 km, with a prevalence of 300-500 m as the most common spacing.

[Figure]

Fig C2. Digitized sounding points from Bangladesh Navy charts (in yellow), the red boxes show the individual chart outlines.

These digitized points do not cover the inland area, which is separately covered by a high-resolution (50 m) inland topography dataset over the south-central part of the delta on Bangladesh side (as explained in L169 old, L182 in revised manuscript), and rest is complemented by SRTM dataset (refer to L171 old, L184 in revised manuscript). The extent of the CEGIS data over inland is shown in Fig C3.

[Figure]

Fig C3. Extent of the 50 m resolution inland topography dataset (in red).

We included Fig. C2 and Fig. C3 as Supplementary figures S1 and S2.

Page 8: Line 173: Is this the whole Bay of Bengal or part of it? Because the latitude extents given in the brackets don't cover the entire BoB. Please check.

Reply: As stated in the manuscript, our domain covers the Bay of Bengal to the North of 11°N. We agree that traditionally, the commonly accepted geographical limit of the Bay of Bengal is rather around 6°N to 8°N. To remove this ambiguity, we reworded "whole Bay of Bengal" to "northern Bay of Bengal". It is now L186 of revised manuscript.

Page 8: Line 175: 250m resolution near the coast is acceptable for the surge computations. However, is it sufficient for the inland inundation computations?

Reply: Given the current level of detail available for the inland topography, we firmly think that 250 m is a well-suited resolution for inland inundation computation. This resolution is not dictated by the scale of the hydrodynamic features, but rather by the resolution of the topographic databases available over our domain. Indeed, the currently available topographic datasets remain rather coarse, in that they do not resolve the sharp man-made physical features – e.g., embankments, roads etc. By combining separate datasets of embankment geometry and using the flexibility of the unstructured grid, we tried to capture the outlines and heights of the embankments in our model grid. As we showed in Figure 8 (Figure 9 in the revised manuscript), these embankments pose the zeroth-order control on the inland inundation, and yet we have limitations of the knowledge of the up-to-date embankment's heights.

However, at a much finer scale, one will think of implementing road-networks, associated water control structures (culverts, bridges etc.) once such high-resolution information becomes available.

Page 8: Line 176: Mentioned here that the model domain and mesh are shown in figure 9, but no figure in the draft shows the mesh and domain. It seems the figure is missed. The domain and mesh figure are important for the readers and hence it should be provided.

Reply: We are sorry for the misunderstanding here; we were referring to Figure 9 of Khan et al. (2019). Our sentence reads correctly in this regard. This being said, we agree with the reviewer that the inclusion of the figure will make it more convincing to the reader. We included the following Figure C4 showing model domain, mesh, and boundary conditions as Figure 4 in the manuscript.

[Figure]

Figure C4. Computational domains and model mesh for SCHISM-WWMIII, as well as model boundary conditions. White arrows on the southern boundary show the forcing with the tidal solution provided by FES2012, and on the northern boundary shows the river discharges. For a hindcast experiment, wave spectra from WW3 are imposed on the southern boundary.

Page 8: Line 178: Mentioned here as 'wave model, is coupled online with SCHISM'. What does it mean? Whether the wave model also uses the same unstructured mesh or it uses the structured mesh? Whether both the models are running at a time (i.e., in parallel)? or running the wave model and then transfer the wave boundary condition to the surge model? These points are to be briefly explained. The given citation Roland et al., 2012 can be used for the complete details.

Reply: By 'coupled online' we meant that two models are fully coupled (and run as a unique executable, coupled at source code level without any external coupler). To avoid ambiguity, we reworded the segment as 'coupled at source code level'.

The wave model uses the same unstructured grid, as mentioned in the following line (L179 old, L192 in the revised manuscript).

The hydrodynamic core and the wave model run sequentially. To reflect this, we revised L183-184 of the old manuscript 'Water level, and current are exchanged among the two models every 30 minutes.' to 'Every 30 minutes of SCHISM runtime, water level and currents are passed to WWM for calculating the evolution of the wave fields. Calculated wave radiation stress, total surface stress, and the wave orbital velocity are passed back to SCHISM before computing the next time step.' (L195-197 of the revised manuscript)

Page 8: Line 200: Given that the blended wind field is used. What is the horizontal resolution of the wind and pressure fields?

Reply: The final resolution of the gridded wind and pressure field is 0.025° (roughly equivalent to 2.8km). We added the following line at the end of L209 (previous manuscript) – 'The final resolution after merging the analytical wind field with the interpolated background GFS fields is 0.025°.' (L225-226 of the revised manuscript)

Following another review comment, we have updated Figure 6 (now Figure 7 in the revised manuscript) with a schematic of the spatial merging of wind and pressure fields from various sources. Considering these two revisions, we also revised corresponding sentences as 'The analytical and background wind fields were first temporally interpolated every 15 minutes and overlaid on the background GFS fields using a distance-varying weighting coefficient to ensure a smooth transition (Figure 6).' (L223-224 of the revised manuscript).

It seems the tide is also included in the computations. What is the spinup time used in the study to get the actual tide levels at the coast?

Reply: Yes, we confirm that the tide is included in the computations, as stated in line 204-207 of the revised manuscript. The spinup time is 2 days or longer in all our numerical simulations (either in hindcast mode, Section 4, or in forecast mode, Section 5). We added the following sentence to make it clear (at L226 of revised manuscript):

"… Lin and Chavas (2012). For all storm surge simulations, a spinup time of 2 days is considered in this study."

What is the source of the buoy data used in the study?

Reply: buoy data used in this study is collected from INCOIS data portal (https://incois.gov.in/portal/datainfo/mb.jsp) during the time of the study which is mentioned in L222 (L240 of the revised manuscript), and the caption of Figure 4 (in revised paper Figure 5). We have made it explicit in the text which now reads - "(the dataset is collected from INCOIS data portal at https://incois.gov.in/portal/datainfo/mb.jsp)".

Figure 4: There is a clear mismatch between observation and modeled total water level at the given locations especially at Angtihara and Tajumuddin. This might be due to the lack of spinup for tide simulation. The reason given here is 'The local bathymetric error and friction parameterization might be the source of the discrepancy'. But the same model used by Krien et al. while using the digitized sounding points has computed the better tide amplitudes. Please check the spinup time. As mentioned if Angtihara is located in a data-scarce location inside Sundarbans mangrove forest, remove the plot.

Reply: The spinup time for the model (including the tides) is of 2 days, which we believe is much more than the typical time needed for a full tidal spin-up (order of 24 h over our domain in our model; not shown). We do not agree that Krien et al. (2016) obtained better tide amplitudes than ours. As seen in our supplement Table S1, their model performance was generally similar to ours, and for the mouth of Meghna - worse than ours. What is more, they did not assess the performance of their model in any of these two stations of Angtihara and Tajumuddin. However, we acknowledge that the plot of Angtihara time-series is perhaps misleading, and removed it in the revised manuscript.

Provide the water level - tide (surge residual) plots (time series) too, to support the statement in the line numbers 246-247, page 11.

Reply: Given the very limited length of observed tide gauge records that were available to us (13 days, including time period with surges), it is unfortunately not possible to operate a meaningful tidal analysis (e.g. Pugh and Woodworth, 2014). Hence it is not feasible to compute any observed surge residual, although we agree it would have improved the clarity of this statement.

**Inundation section:** The methodology is to be clearer. Though Lewis et al. is cited for the details, a brief description is required here. Whether the model mesh extends on to the land or not? if so up to what extent (i.e., up to which topography contour)? or whether the water level values at the coast are used and extrapolated the inundation extents?

Reply: The model extent of the model mesh is now shown in a new Figure 4 (Figure C4). The model mesh does extend over land, up to beyond the topography contour of 5 m above MSL.

Regarding the calculation methodology, it seems there is a misunderstanding as we did not cite Lewis et al. 2013 "for details" relevant to our own modeling framework. Indeed, unlike Lewis et al. 2013, inundation is modelled seamlessly in SCHISM, considering the same hydrodynamics as estuaries or ocean. To clarify the methodology and avoid misunderstanding we added the following line at L356 (L370 in the revised manuscript) –

'… (Lewis et al. 2013). In our modelling framework, the inland inundation is calculated seamlessly by SCHISM, solving the same hydrodynamics over the model domain, thanks to the combination of cross-scale unstructured-grid and an efficient wetting-drying algorithm. While the recent improvement…'

**References**

Pugh, D., & Woodworth, P. (2014). *Sea-level science: understanding tides, surges, tsunamis and mean sea-level changes*. Cambridge University Press.

**Replies to comments of Reviewer #2**

This paper presents an interesting strategy to address surge and inundation in the bay of Bengal, based on publicly available model and forcings. The quality of the results is much larger than the previous efforts and a detailed analysis on process dominance is presented. The proposed strategy, although apparently customized to this site, is interesting and worth publishing. Below I have outline major and minor issues to be solved before approval, some of them requiring new runs. Furthermore, the paper is extremely long (sometimes it looks more a report than a scientific paper) and lacks now and then a clear presentation direction. The paper should thus be reviewer for conciseness and easy of reading.

I consider the paper to be accepted with major reviews.

Reply: We thank the reviewer for her/his insightful comments. Following the comments, we operated removal and revision of several paragraphs as explained hereafter, in order to reduce the length and to increase the clarity as well as the readability of the manuscript.

We reorganized the Appendix of the previous version to Supplementary materials to keep the main manuscript concise. The Appendix Table 2 showing the dataset is now inserted in the text as Table 1.

The final manuscript amounts to 19 printed page in the final journal format (29 pages in the manuscript format).

Abstract

"Despite recent advancements, the complex morphology and hydrodynamics of this large delta and the associated modelling computational costs impede the storm surge forecasting in this highly vulnerable region." – Nowadays, super computers are available to perform forecasts of much larger and more complex domains, integrated with atmospheric models. The author should review this sentence maybe focusing on the quality of the forecasts and the necessary grid refinements and process-knowledge for high quality results.

Reply: We agree with the suggested changes. We revised as following at L1-3 of the revised manuscript:

"Despite recent advancements, the complex morphology and hydrodynamics of this large delta and the associated modelling complexity impede accurate storm surge forecasting in this highly vulnerable region."

"This article shows the proof of the maturity of our framework for operational implementation, which can particularly improve the quality of localized forecast for effective decision-making" –

Is the framework generic or only applicable to the bay of Bengal? The authors should clarify this issue at the abstract.

Reply: Our modelling system, as well as the forecasting procedure demonstrated in this study, are generic enough to be directly applicable to other regions. We thank the reviewer for suggesting this improvement. We updated the line L8-9 (L7-10 of the revised manuscript) as follows:

"This article shows the proof of the maturity of our framework for operational implementation, which can particularly improve the quality of localized forecast for effective decision-making over the Bengal delta shorelines, as well as over other similar cyclone-prone regions."

Line 36 "global weather and forecasting system" – a word is missing of the "the" word needs to be removed.

Reply: We suspect a misunderstanding here. To us, our sentence is grammatically correct (L37 of the revised manuscript):

" Over the last decades, global weather and forecasting systems have advanced significantly."

Lines 44-45 "Nowadays, operational surge forecasting systems typically run on high-performance computing systems, either on a scheduled basis or triggered on-demand during an event (Khalid and Ferreira, 2020)" – The authors should include other references of such system, either applicable at a specific site or of generic application.

Reply: We agree. We revised the text with the following recent references: Loftis et al., 2019; Oliveira et al., 2020. The updated sentences corresponds to L45-47 of the revised manuscript.

Lines 50-52 "Storm surge forecasts have shown their potential to better target the evacuation decision, to optimize early-engineering preparations, and to improve the efficiency of the allocation of the resources (Glahn et al., 2009; Lazo and Waldman, 2011)." – again an updated and more comprehensive review is missing, along with the identification of what are the major challenges in developing and keeping in operational mode this type of systems.

Reply:  We have revised the segment to (L51-59 of the revised manuscript)–

"Storm surge forecasts have shown their potential to better target the evacuation decision, to optimize early-engineering preparations, and to improve the efficiency of the allocation of the resources (Glahn et al. 2009; Lazo and Waldman, 2011, Munroe 2018). Availability of a spatially-distributed forecast of storm surge flooding can further increase the fluidity of communication toward the public (Lazo et al. 2015). Keeping in mind the cyclonic surge hazards over the

densely populated Bengal delta, having a reliable real-time operational forecast system in the region would be extremely valuable and would address a societal demand (Ahsan et al. 2020). The major challenges in operating and maintaining such systems are manifold for the Bengal delta, including lack of expertise, limitations of funding resources to operate and maintain necessary infrastructure and dataset, as well as availability of reliable modelling systems in operational mode (Roy et al. 2015)."

Lines 58-60 "In the past decade, unstructured-grid modelling systems are getting more and more popular due to their efficiency in resolving the topographic features and their reduced computational cost compared to structured-grid equivalents (Ji et al., 2009; Lane et al., 2009; Melton et al., 2009)." – all these references are not from the last decade. Part of them do not address operational forecast systems. There are several examples of unstructured grid forecast system in operation, some recent some in operation for over a decade. The authors should review carefully the state-of-the-art and improve the current text.

Reply: In this particular sentence, only the unstructured-grid modelling aspect is focused, irrespective of forecasting context. However, we agree with the reviewer regarding the state-of-the-art forecasting systems and revised by adding the following references - Fortunato et al. 2017, Khalid and Ferreira 2020. These changes are now in L64-66 of the revised manuscript reading -

"Over the past decades, unstructured-grid modelling systems are getting more and more popular due to their efficiency in resolving the topographic features and their reduced computational cost compared to structured-grid equivalents (Ji et al., 2009; Lane et al., 2009; Melton et al., 2009; Fortunato et al., 2017; Khalid and Ferreira, 2020)."

Both of these references are already cited elsewhere in the current manuscript.

Line 100 "Due to this interaction, the highest surge is obtained for a storm making landfall around 2 hours before the high tide." – a detailed explanation (or references explaining it) is needed. Is it associated with the specifics of the geometric/bathymetric characteristics of the bay or generic? Is it tidal amplitude dependent?

Reply: To clarify, we propose to add references to Krien et al. 2017 and Antony et al. 2020 as references to numerical investigation, and Antony et al. 2013 for simplified mathematical insight. The tide-surge interaction is a generic phenomenon, which evolves dynamically, and which is prominent at locations with shallow submarine zones and/or strong tidal currents. These past studies suggest that the tide-surge interaction is generic and depends on the tidal amplitude and tidal phase, which are primarily controlled by the regional bathymetry, regional coastline geometry, and the ocean bottom roughness. Considering momentum equations, tide-surge interactions can be explained by terms that depend on the water depth (surface and

bottom stress) and terms that depend on current velocity (advection, acceleration, and bottom stress). The "non-linear" interaction is thus dependent on these regional features and does not have a one-to-one relationship with tidal amplitude.

The statement in L100, was a rough estimate when the highest water level can be observed based on previous experiments (e.g. Krien et al. 2017). In the English channel, for instance, the maximum surge height is reached 4-6 hours before high tide (Idier et al. 2012). However, no estimate is yet available across the Bengal delta shorelines. Due to above-mentioned contributing processes, the response is expected to vary along the coastlines and inside the estuaries.

We updated the line as following (L110-111 of the revised manuscript) –

"Due to this interaction, the highest surge (water level – tide) does not coincide with high tide as shown from observations (Antony et al. 2013) and numerical modelling (Krien et al. 2017, Antony et al. 2020)."

Line 165-169 "Our bathymetric dataset is a blend of two digitized sounding datasets in the nearshore zone – one from navigational charts produced by Bangladesh Navy, and another being a bathymetry of the Hooghly estuary provided by IWAI (Inland Waterways Authority of India)"- are there any common areas between the two sources? if yes, what was the combination procedure? If not, substitute "blend" by "combination". How old is the data?

"The river bathymetry is composed of a set of cross-sections obtained from the Bangladesh Water Development Board (BWDB)." – what is the spacing between profiles? How old is the data?

Reply: There are common areas between the sounding dataset. As they are point-wise sounding referenced to an uniform datum, they are combined as is. The river cross-section bathymetry are blended with priorities to the sounding from the navigational charts.

As suggested, we adopt "combination" to better represent the characteristic of the merged dataset. Depending on the locations, the data are 7 to 20 years old.

The spacing of the sections of the river profiles is typically 10-30Km. However, these cross-sections points are interpolated using dedicated interpolation tool (HEC-RAS + GIS) to about 300 m spacing before combining with the rest of the bathymetric dataset. Depending on the regions, the cross-sections are 10 to 15 years old.

We revised L165-169 as following (L177-181 of the revised manuscript):

"Our bathymetric dataset is a combination of two digitized sounding datasets in the nearshore zone – one from navigational charts produced by Bangladesh Navy, and another being a bathymetry of the Hooghly estuary provided by IWAI (Inland Waterways Authority of India).

Depending on the sounding points, these observations are 7 to 20 years old. The river bathymetry is composed of a set of cross-sections obtained from the Bangladesh Water Development Board (BWDB), which is further interpolated at about 300m resolution using dedicated 1D river modelling tool and GIS techniques."

Lines 194-195 "At each of the upstream river open boundaries of Ganges, Brahmaputra, Hooghly, and Karnaphuli, we implemented a discharge boundary condition" – what is the source of the discharge values?

Reply: For the benchmark tidal simulation, for Ganges and Brahmaputra, the discharge is estimated from observed water level provided by BWDB and appropriate rating curves. For Hooghly and Karnaphuli, climatological discharge is taken from Mukhopadhyay et al. 2006 and Chowdhury et al. 2012 respectively. Finally, for the storm surge simulations in this study, a climatologic discharge is estimated from the discharge timeseries at Ganges and Brahmaputra.

We revised L194-195 (L208-211 of revised manuscript)–

"At upstream river open boundaries of Ganges and Brahmaputra, a discharge time series from BWDB is forced for the benchmark tidal simulation, and a climatologic discharge timeseries is applied for storm surge simulations during Amphan. A discharge climatology is applied at Hooghly (Mukhopadhyay et al. 2006) and Karnaphuli (Chowdhury et al. 2012)."

Fig 4 – the map is unreadable. Place it at a larger scale.

Reply: We revised the layout of Figure 4 (revised Figure 5), making it full-width, as in Figure C1. Additionally, one of the tide gauge station (Angthihara) is removed following a comment from another reviewer.

[Figure]

Figure C1. Comparison of simulated (in orange) and observed (in blue) water level, significant wave height (SWH) and mean wave period. (a) The map shows the along-track bias in SWH compared to the one calculated from Sentinel 3B altimeter overpass at 2020-05-18 1603Z. Bottom panel shows the modelled SWH and means wave period (orange line) compared to buoy observations (blue dots) at BD08 (b-c) and BD11 (d-e) provided by INCOIS. Comparison between observed (blue dots) and modelled (orange line) water level at the station locations – (f) Galachipa, (g) Kuakata, (h) Tajumuddin, (i) Chittagong, and (j) Visakhapatnam. Green dashed lines in (f)-(k) indicates the modelled tidal water level. Location of the buoys and the water level gauges are shown in (a). The vertical red lines in water level plots indicate the time of landfall.

Lines 200 to 209 this paper aims at evaluating a procedure (framework is not adequate in this context) to forecast storm surges and evaluates the procedure using a past event. However, when running an operational model, reanalysis are not available. Therefore the quality of the model should be evaluated with a past event (so data is available) but for a run under operational forecast conditions. The analysis of fig 4 should therefore by re-done under these conditions.

Reply: We do not believe it would be relevant to include the hindcast and model validation of another different event, for the sake of the conciseness of an already long paper. Indeed, we already tested the modeling framework in hindcast mode, for several past events with a configuration very similar to the one used in the present study (Krien et al. 2017). Essentially, our objective, as stated in the introduction (L. 80-81 of the old, L90-92 of the revised manuscript), is to assess the model in an operational mode. Still, we understand the need for validation of the present model. The objective of Section 4.3 and previously Figure 4 (revised Figure 5) is precisely to evaluate the model. Thus the presented hindcast experiment is performed after-event. These results can be considered as the best we can achieve just after the cyclone has passed.

Lines 213 and following: why is the comparison limited to the storm path? The data is available in the whole domain.

Reply: The significant wave height is estimated from available along-track altimetric measurements. Thus, the comparison is also along the track of the altimetric record over the whole domain. Given the repeat cycle of Sentinel 3 (27 days), we consider ourselves lucky in this regard to have Sentinel 3B flying nearby the cyclone track during a cyclone which lasted about 7 days. Additionally, outside of cyclone track region, the wave heights were small (Hs < 2m) (previously Figure 4, current Figure 5 - beginning time of the BD08, and ending time of BD11), which is also well reproduced by the model.

Lines 274 and following: this text belongs in the introduction. Remove.

Reply: Agreeing to this point, we removed L274-L280 in the old manuscript "A realistic… … … evacuation orders". To maintain the continuity, we add the following sentence – "In this section, a near real-time storm surge forecasting scheme is presented using publicly available atmospheric forcing dataset."

Lines 282-284: "We communicated the results to Bangladesh local government authority through personal communications, as well as to the scientific community through social media."

– this sentence is off context and has no scientific link with the remaining of the chapter. It should be moved to other parts of the paper (introduction?)

Reply: We agree and shifted these lines to the conclusions (L441 of the revised manuscript).

Fig 6 only deals with time. It should be improved with a plot on space definition of forcings. The use of the "blend" should be reviewed.

Reply: We agree. We revised Figure 6 (current manuscript, 7 in the revised manuscript) with the inclusion of the spatial definition of the forcing, as below (Figure C2). We confirm "blend" is appropriate in this case as overlapping regions are treated with various weighting factors:

1. From the storm center to 3xRm: only analytical field from JTWC/HWRF
2. From 3xRm to 10xRm: linear transition from analytical fields to GFS fields
3. Outside of 10xRm: only GFS field

[Figure]

[Figure]

Figure C2. (a) Temporal combination scheme of the JTWC, GFS, and HWRF forecasts for each 6-hourly storm surge forecast epoch. (b) Spatial blending of the analytical and GFS wind and pressure field.

Page 13 – the proposed methodology seems too linked with the specific physics of the site and of this particular event. Small variations of the coupling should be tested and compared with data.

Reply: We disagree with the Reviewer on this point as all the datasets used in Section 5.1 (Page 13 of the previous, 14 of the revised manuscript) are global and publicly accessible, thus not specific to the particular site or the event. Also, the modelling system (SCHISM-WWM) is used to hindcast storm surges and flooding in many regions of the world (e.g. Gulf of Mexico, East US coast, Western Europe, etc.), with model configurations that differ little compared to the one used in this study.

Varying the forcing strategy (assuming this is what the Reviewer means by "coupling") is alas not possible, as HWRF forecast is the sole product operationally available over the domain of our study forecasting the essential wind and pressure fields information.

One other forcing option is relying only on publicly available forecast fields (e.g. GFS at 0.25° resolution). In such resolution, the cyclone core is typically not well defined and results in a weaker wind and pressure field compared to merging technique as discussed in a later comment reply.

Lines 315 and following: errors are necessary for the forecast runs. The discussion is very weakly supported without them.

Reply: We agree. We assessed the errors of the various forecast runs with respect to the hindcast run, in terms of maximum surge level simulated. We included the results in a modified version of Figure 7 (current manuscript, 8 in revised manuscript), as shown here (Fig. C3). We revised the text at L317 (L329 of the revised manuscript) as following:

"To substantiate the gradual increase in the quality of the forecasts, we compared the maximum surge level predicted at the various forecast ranges (T-60 hours, T-36 hours, T-12 hours) with the hindcast experiment, along a line encompassing the near-shore delta (segment displayed on Figure 6). The results show that, in the T-60h forecast, the location of maximum surge appears offset eastward, by as much as 150 km. The magnitude of the maximum surge is also poorly predicted, with an overestimation of about 3 m. In the T-36h forecast, the location of the maximum surge appears largely corrected, but the magnitude of the peak remains overestimated by about 3 m. In the T-12h forecast, both the location and magnitude of the peak surge are in relatively good agreement with the hindcast. Overall, along this landfalling coastal section, the standard error of the maximum surge level amounts to 2.06 m, 1.73 m, and 0.66 m for the T-60h, T-36h, and T-12h forecast, respectively."

[Figure]

Figure C3. Maximum surge (a, c, e) and elevation (b, d, f) evolution for forecast initiated at (a-b) T-60 hours (2020-05-18 0000Z), (c-d) T-36 hours (2020-05-19 0000Z), (e-f) T-12 (2020-05-20 0000Z) hours before landfall. (g) Comparison of maximum surge level extracted along the section shown in white line in (a, c, e). Hindcast results (BEST) is extracted along the same line shown in Figure 6(b).

Current Figure 5 (will become Figure 6 in the revision) is shown in Figure C4.

[Figure]

Figure C4. Hindcast of (a) maximum water level, (b) maximum surge (c) wave setup/setdown (d) maximum non-linear interaction between tide and surge. For (a), for the areas above mean sea level, the water level is converted to water level above the topography for consistency. The inset maps show a close-up (75 km × 45 km) of the landfall region. The black dashed line shown in (b) is the segment for error analysis of the forecast experiment in Section 5.

Lines 327 until the end of page 15 belong in the introduction as motivation for this study.

Reply: We agree and moved the part from L327 to L332 (of the previous manuscript) to introduction, to replace the part from L71 to L74 of the previous manuscript (L77-79 of the revised manuscript).

Lines 344: "The best way to avoid the error from the analytical wind field might be not using these formulations and rely on the full-fledged atmospheric forecasts". In spite of the limitations of existing atmospheric forecasts, this sentence and the next ones would be better supported with a simulation just based on the available forecasts. I suggest the author to repeat the simulation without the analytical model and evaluate the differences.

Reply: We performed a dedicated sensitivity test, with a hindcast simulation forced only with the GFS data (accumulations of forecasts for each 6 hours cycles over the cyclone period). Compared to our nominal forcing strategy, the central pressure along the cyclone trajectory is much overestimated in the GFS data, by about 20hPa on the day of landfall. Similarly, the wind is slightly weaker in the GFS data (typically by about 5 to 10 m/s) (see Fig. C5). As a result, the surge height is largely underestimated in the region of the cyclone landfall typically by 50% (Fig C6). As this conclusion was largely expected based on the numerous studies published on this issue so far (e.g. Zhang et al. 2013), we believe it is not relevant to include this in the manuscript.

[Figure]

Fig C5. Comparison between JTWC and GFS maximum wind speed, and central pressure.

[Figure]

Fig C6. Maximum water level for (a) nominal forcing strategy used in this study combining analytical and GFS fields, (b) only GFS fields.

Fig 8 – why are the inundation patches for the hindcast experiment and not for the forecast runs? I suggest that those are included too, with another figure.

Reply: The objective of Figure 8 (Figure 9 in the revised manuscript) is not to assess the quality of the forecast, rather to point towards the challenges in modelling the inundation. As suggested by the reviewer, we have added the inundation patches for the forecast experiments as a separate figure (Figure C7). They show that the flooding of the forecast runs is largely in line with the hindcast experiment, in particular over the various sub-regions discussed in the corresponding section (Section 6.3). Thus, to keep the main manuscript lean and focused, we added them as supplementary figure in the revised manuscript.

[Figure]

Figure C7. Same as Figure 8 for forecasts at T-60, T-36, and T-12 hours.

Lines 400-401 – refer to table A2. In table A2, correct the 2$_{nd}$ link as it is not a public link.

Reply: We have revised and moved Table A2 to Table 1 in the main text.

We are sorry for the misplacement of the link. The correct link is https://www.emc.ncep.noaa.gov/gc_wmb/vxt/HWRF/index.php

References:

Antony, C., Unnikrishnan, A., Krien, Y., Murty, P., Samiksha, S., and Islam, A.: Tide–surge interaction at the head of the Bay of Bengal during Cyclone Aila, Regional Studies in Marine Science, 35, 101 133, https://doi.org/10.1016/j.rsma.2020.101133, 2020.

Ahsan, M.N., Khatun, A., Islam, M.S., Vink, K., Ohara, M. and Fakhruddin, B.S., 2020. Preferences for improved early warning services among coastal communities at risk in cyclone prone south-west region of Bangladesh. Progress in Disaster Science, 5, p.100065.

Chowdhury, M.A.M., Al Rahim, M., 2012. A proposal on new scheduling of turbine discharge at kaptai hydro-electric power plant to avoid the wastage of water due to overflow in the dam. In: 2012 7th International Conference on Electrical and Computer Engineering. IEEE, pp. 758–762.

Fortunato, A.B., Oliveira, A., Rogeiro, J., Tavares da Costa, R., Gomes, J.L., Li, K., de Jesus, G., Freire, P., Rilo, A., Mendes, A., Rodrigues, M., Azevedo, A., 2017. Operational forecast framework applied to extreme sea levels at regional and local scales. Journal of Operational Oceanography, 10 (1), pp. 1-15.

Lazo, J. K., Bostrom, A., Morss, R. E., Demuth, J. L., and Lazrus, H.: Factors Affecting Hurricane Evacuation Intentions, Risk Analysis, 35, 1837–1857, https://doi.org/10.1111/risa.12407, 2015.

Loftis, J.D., Mitchell, M., Schatt, D., Forrest, D.R., Wang, H.V., Mayfield, D., Stiles, W.A., 2019. Validating an operational flood forecast model using citizen science in Hampton Roads, VA, USA. J. Mar. Sci. Eng. 7, 242. https://doi.org/10.3390/ jmse7080242.

Munroe, R., Montz, B. and Curtis, S., 2018. Getting more out of storm surge forecasts: emergency support personnel needs in North Carolina. Weather, climate, and society, 10(4), pp.813-820.

Mukhopadhyay, S.K., Biswas, H., De, T.K., Jana, T.K., 2006. Fluxes of nutrients from the tropical river hooghly at the land–ocean boundary of sundarbans, NE coast of Bay of bengal, India. J. Mar. Syst. 62 (1–2), 9–21. https://doi.org/10.1016/j.jmarsys.2006.03.004.

Oliveira, A., Fortunato, A., Rogeiro, J., Teixeira, J., Azevedo, A., Lavaud, L., Bertin, X., Gomes, J., David, M., Pina, J., Rodrigues, M., and Lopes, P.: OPENCoastS: An open-access service for the automatic generation of coastal forecast systems, Environmental Modelling & Software, 124, 104 585, https://doi.org/10.1016/j.envsoft.2019.104585,2020.

Roy, C., Sarkar, S. K., Aberg, J., and Kovordanyi, R.: The current cyclone early warning system in Bangladesh: Providers and receivers views, International Journal of Disaster Risk Reduction, 12, 285–299, https://doi.org/10.1016/j.ijdrr.2015.02.004, publisher: Elsevier BV, 2015

Zhang, H., and J. Sheng, 2013. Estimation of extreme sea levels over the eastern continental shelf of North America. Journal of Geophysical Research Oceans. 118, 6253–6273, doi:10.1002/2013JC009160